# AUGMENTATION-AWARE SELF-SUPERVISED LEARNING WITH CONDITIONED PROJECTOR

## ABSTRACT

Self-supervised learning (SSL) is a powerful technique for learning robust representations from unlabeled data. By learning to remain invariant to applied data augmentations, methods such as SimCLR and MoCo are able to reach quality on par with supervised approaches. However, this invariance may be harmful to solving some downstream tasks which depend on traits affected by augmentations used during pretraining, such as color. In this paper, we propose to foster sensitivity to such characteristics in the representation space by modifying the projector network, a common component of self-supervised architectures. Specifically, we supplement the projector with information about augmentations applied to images. In order for the projector to take advantage of this auxiliary conditioning when solving the SSL task, the feature extractor learns to preserve the augmentation information in its representations. Our approach, coined Conditional Augmentation-aware Self-supervised Learning (CASSLE), is directly applicable to typical joint-embedding SSL methods regardless of their objective functions. Moreover, it does not require major changes in the network architecture or prior knowledge of downstream tasks. In addition to an analysis of sensitivity towards different data augmentations, we conduct a series of experiments, which show that CASSLE improves over various SSL methods, reaching state-of-the-art performance in multiple downstream tasks.[1]

## 1 INTRODUCTION

Artificial neural networks have proven to be a successful family of models in several domains, including, but not limited to, computer vision (He et al., 2016), natural language processing (Brown et al., 2020), and solving problems at the human level with reinforcement learning (Mnih et al., 2015). This success is attributed largely to their ability to learn useful feature representations (Goodfellow et al., 2016) without additional effort for input signals preparation. However, training large deep learning models requires extensive amounts of data, which can be costly to prepare, especially when human annotation is needed (Bai et al., 2021; Kim et al., 2022).

High-quality image representations can be acquired without relying on explicitly labeled data by utilizing self-supervised learning (SSL). A self-supervised model is trained once on a large dataset without labels and then transferred to different downstream tasks. Initially, self-supervised methods addressed well-defined pretext tasks, such as predicting rotation (Gidaris et al., 2018) or determining patch position (Doersch et al., 2015). Recent studies in SSL proposed contrastive methods of learning representations that remain invariant when subjected to various data augmentations (He et al., 2020; Chen et al., 2020a; Chen & He, 2021) leading to impressive results that have greatly diminished the disparity with representations learned in a supervised way (Caron et al., 2021).

Nevertheless, contrastive methods may perform poorly when a particular downstream task relies on features affected by augmentation (Xiao et al., 2021). For example, color jittering can result in a representation space invariant to color shifts, which would be detrimental to the task of flower classification (see Figure 1). Without prior knowledge of possible downstream tasks, this effect is hard to mitigate in contrastive learning (Tian et al., 2020; Xiao et al., 2021). Solutions for retaining

---

[1]We share our anonymized codebase at `https://anonymous.4open.science/r/CASSLE-037C`.

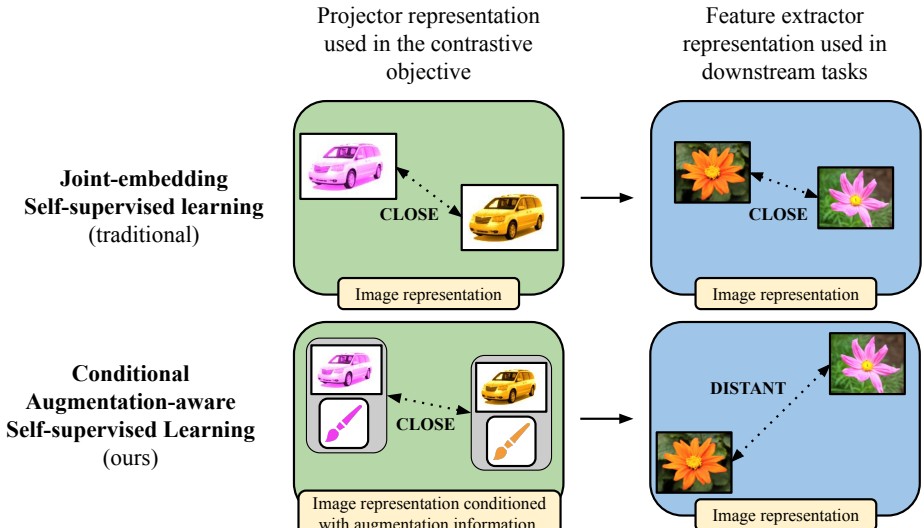

Figure 1: In the traditional self-supervised setting, contrastive loss minimization pulls the representations of augmented image views closer in the latent space of the projector (left). This may also reduce the distance between their feature extractor representations (right). Thus, the representation becomes invariant to augmentation-induced perturbations, which may hinder the performance on downstream tasks. In contrast, the self-supervised objective of CASSLE draws together joint representations of images and their augmentations in the projector space (bottom row). By conditioning the projector with augmentation information, image representations retain more sensitivity to perturbations in the feature extractor space. This proves to be beneficial when solving downstream tasks.

information about used data augmentations in the feature extractor representation include forcing it explicitly with a modified training scheme (Xiao et al., 2021; Lee et al., 2021; Xie et al., 2022), or by preparing a feature extractor to be adapted to a specific downstream task, e.g., with hypernetworks (Chavhan et al., 2023). However, these approaches often involve significant modifications either to the contrastive model architecture (Xiao et al., 2021), training procedure (Lee et al., 2021; Xie et al., 2022), or costly training of additional models (Chavhan et al., 2023).

In this work, we propose a new method called **C**onditional **A**ugmentation-aware **S**elf-**s**upervised **Le**arning (CASSLE) that mitigates augmentation invariance of representation without neither major changes in network architecture or modifications to the self-supervised training objective. We propose to use the augmentation information during the SSL training as additional conditioning for the projector network. This encourages the feature extractor network to retain information about augmented image features in its representation. CASSLE can be applied to any joint-embedding SSL method regardless of its objective (Chen et al., 2020b;a; Chen & He, 2021; Zbontar et al., 2021; Chen et al., 2021b). The outcome is a general-purpose, augmentation-aware encoder that can be directly used for any downstream task. CASSLE presents improved results in comparison to other augmentation-aware SSL methods, improving transferability to downstream tasks where invariance of the model representation for specific data changes could be harmful.

**The main contributions of our work are threefold:**

- We propose a simple yet effective method for Conditional Augmentation-aware Self-supervised Learning (CASSLE). Using our conditioned projector enables preserving more information about augmentations in representations than in existing methods.

- CASSLE is a general modification that can be directly applied to existing joint-embedding SSL approaches without introducing additional objectives and major changes in the network architecture.

- In a series of experiments we demonstrate that CASSLE reaches state-of-the-art performance with different SSL methods for robust representation learning and improves upon the performance of previous augmentation-aware approaches. Furthermore, our analysis indicates that CASSLE learns representations with increased augmentation sensitivity compared to other approaches.

## 2 RELATED WORK

**Self-supervised learning** (SSL) is a paradigm of learning representations from unlabeled data that can later be used for downstream tasks defined by human annotations (Albelwi, 2022; Balestriero et al., 2023). Despite learning artificial *pretext tasks*, instead of data-defined ones, SSL models have achieved tremendous success in a plethora of domains (Devlin et al., 2019; Wickstrøm et al., 2022; Schiappa et al., 2022; Bengar et al., 2021). This includes computer vision, where a variety of pretext tasks has been proposed (Doersch et al., 2015; Zhang et al., 2016; Noroozi & Favaro, 2016; Gidaris et al., 2018). However, arguably the most prominent and successful SSL technique to emerge in recent years is the training of joint-embedding models for augmentation invariance (Becker & Hinton, 1992; van den Oord et al., 2019), defined by objectives such as contrastive InfoNCE loss (He et al., 2020; Chen et al., 2020a;b), self-distillation (Grill et al., 2020; Chen et al., 2020a; Oquab et al., 2023) or Canonical Correlation Analysis (Caron et al., 2020; Zbontar et al., 2021; Bardes et al., 2022). Those objectives are often collectively referred to as *contrastive objectives* (Tian, 2022; Balestriero et al., 2023). A common component of joint-embedding architectures is the *projector network*, which maps representations of the feature extractor into the space where the contrastive objective is imposed (Chen et al., 2020a;b). The usefulness of the projector has been explained through the lens of transfer learning, where it is often better to transfer intermediate network representations, to reduce the biases from the pretraining task (Yosinski et al., 2014; Bordes et al., 2023). The projector also helps to mitigate the noisy data augmentations and enforces some degree of pairwise independence of image features (Balestriero et al., 2023; Mialon et al., 2023).

**Augmentation invariance of self-supervised models** is a natural consequence of training them with contrastive objectives, as SSL methods are prone to suppressing features that are not useful for optimizing the contrastive objectives (Chen et al., 2021a; Robinson et al., 2021). While a common set of augmentations demonstrated to typically work well on natural images in SSL has been established in the literature (He et al., 2020; Chen et al., 2020a; Chen & He, 2021; Caron et al., 2020; Zini et al., 2023), the optimal choice of augmentations varies between specific tasks (Tian et al., 2020; Ericsson et al., 2022). (Xiao et al., 2021) find that augmentation invariance can hinder the performance of the model on downstream tasks that require attention to precisely those traits that it had been previously trained to be invariant to. On the other hand, (Zhang et al., 2019) observe that the objective of predicting augmentation parameters can be in itself a useful pretext task for SSL. Those works inspired several techniques of retaining augmentation-specific information in joint-embedding models, such as projectors sensitive to different augmentation types (Xiao et al., 2021; Ericsson et al., 2022), adding an objective of explicit prediction of augmentation parameters (Lee et al., 2021), , as well as task-specific pretraining (Raghu et al., 2021; Wagner et al., 2022). The above approaches produce general-purpose feature extractors that can be transferred to downstream tasks without further tuning of their parameters. However, they often involve complex modifications either to the SSL model architecture (Xiao et al., 2021), training procedure (Lee et al., 2021; Xie et al., 2022), or simply tedious task-specific pretraining (Wagner et al., 2022). Another line of work proposes to train Hypernetworks (Ha et al., 2017) which produce feature extractors invariant to chosen subsets of augmentations – a more elastic, but considerably harder to train approach (Chavhan et al., 2023). Several works have proposed fostering the equivariance of representations to data transformations using augmentation information. Xie et al. (2022) modulate the contrastive objective with augmentation strength, Bhardwaj et al. (2023) use augmentation information as a signal for equivariance regularization in the supervised setting, whereas Garrido et al. (2023) extend the VicReg Bardes et al. (2022) objective with a predictor whose parameters are generated from augmentation information by a hypernetwork Ha et al. (2017). Compared to those works, we do not make any modifications to the objective functions of the extended approaches and evaluate CASSLE on a wide range of joint-embedding approaches and conditioning techniques. Following (Xiao et al., 2021; Lee et al., 2021), we produce a general-purpose feature extractor and utilize augmentation information similarly to (Zhang et al., 2019; Lee et al., 2021; Chavhan et al., 2023). Contrary to the above methods, we inject the information about the applied augmentations directly into the projector and make no modification either to the contrastive objective or the feature extractor.

## 3 METHOD

In this section, we present our approach, **C**onditional **A**ugmentation-aware **S**elf-**s**upervised **le**arning (CASSLE). Section 3.1 provides background on joint-embedding self-supervised methods and their

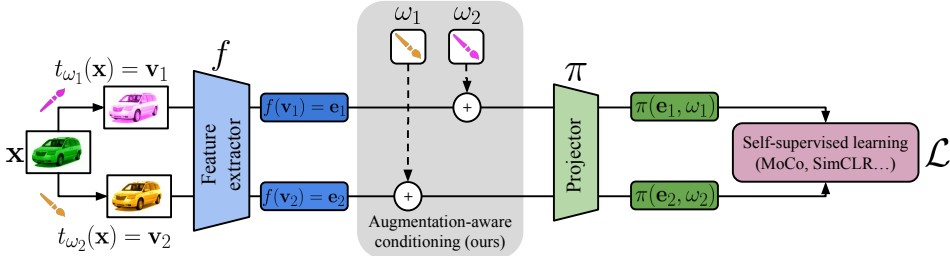

Figure 2: Overview of CASSLE. We extend the typical self-supervised learning approaches by incorporating the information of augmentations applied to images into the projector network. In CASSLE, the SSL objective is thus imposed on joint representations of images and the augmentations that had been applied to them. This way, CASSLE enables the feature extractor to be more aware of augmentations than the methods that do not condition the projector network.

limitations. Section 3.2 explains the essence of CASSLE and how it leverages augmentation information to improve the quality of learned representations. Section 3.3 details the practical implementation of CASSLE's conditioning mechanism.

### 3.1 PRELIMINARIES

A typical contrastive framework used in self-supervised learning consists of an augmentation function $t_\omega$ and two networks: feature extractor $f$ and projector $\pi$. Let $\mathbf{v}_1 = t_{\omega_1}(\mathbf{x}), \mathbf{v}_2 = t_{\omega_2}(\mathbf{x})$ be two augmentations of a sample $\mathbf{x} \sim X$ parameterized by $\omega_1, \omega_2 \sim \Omega$. The feature extractor maps them into the embedding space, which is the representation used in downstream tasks. To make the representation invariant to data augmentations, $\mathbf{e}_1 = f(\mathbf{v}_1)$ is forced to be similar to $\mathbf{e}_2 = f(\mathbf{v}_2)^2$. However, instead of imposing similarity constraints directly on the embedding space of $f$, a projector $\pi$ transforms the embeddings into target space where the contrastive loss $\mathcal{L}$ is applied. This trick, known as *Guillotine Regularization*, helps the feature extractor to better generalize to downstream tasks, due to $f$ not being directly affected by $\mathcal{L}$ (Yosinski et al., 2014; Chen et al., 2020a;b; Bordes et al., 2023).

Minimizing $\mathcal{L}(\pi(\mathbf{e}_1), \pi(\mathbf{e}_2))$ directly leads to reducing the distance between embeddings $\pi(\mathbf{e}_1)$ and $\pi(\mathbf{e}_2)$. However, $\mathcal{L}$ still indirectly encourages intermediate network representations (including the output of the feature extractor $f$) to also conform to the contrastive objective to some extent. As a result, the feature extractor tends to erase the information about augmentation from its output representation. This behavior may however be detrimental for certain downstream tasks (see Figures 1 and 4), which rely on features affected by augmentations. For instance, learning invariance to color jittering through standard contrastive methods may lead to degraded performance on the downstream task of flower recognition, which is not a color-invariant task (Tian et al., 2020; Xiao et al., 2021). Therefore, the success of typical SSL approaches depends critically on a careful choice of augmentations used for model pretraining (Chen et al., 2020a; Tian et al., 2020).

### 3.2 CASSLE

To overcome the above limitations of SSL, we facilitate the feature extractor to encode the information about augmentations in its output representation. In consequence, the obtained representation will be more informative for downstream tasks that depend on features modified by augmentations.

CASSLE achieves this goal by conditioning the projector $\pi$ on the parameters of augmentations used to perturb the input image. Specifically, we modify $\pi$ so that apart from embedding $\mathbf{e}$, it also receives augmentation information $\omega$ and projects their joint representation into the space where the objective $\mathcal{L}$ is imposed. We do not alter the $\mathcal{L}$ itself; instead, training relies on minimizing the contrastive loss $\mathcal{L}$ between $\pi(\mathbf{e}_1|\omega_1)$ and $\pi(\mathbf{e}_2|\omega_2)$. Thus, $\pi$ learns to draw $\mathbf{e}_1$ and $\mathbf{e}_2$ together in its representation space *on condition of $\omega_1$ and $\omega_2$*. We visualize the architecture of CASSLE in Figure 2.

---

[2]While contrastive objectives such as InfoNCE (van den Oord et al., 2019) regularize the representation using negative pairs, we omit them from our notation for the sake of brevity.

We provide a rationale for why CASSLE preserves information about augmented features in the representation space. Since augmentation information vectors $\omega$ do not carry any information about source images $\mathbf{x}$, their usefulness during pretraining could be explained only by using knowledge of transformations $t_\omega$ that had been applied to $\mathbf{x}$ to form views $\mathbf{v}$. However, for such knowledge to be acted upon, features affected by $t_\omega$ must be preserved in the feature extractor representation $f(\mathbf{v})$.

Let us assume the opposite, that $\omega$ is not useful for CASSLE to solve the task defined by $\mathcal{L}$. If this is the case, then for any $\omega_3 \sim \Omega$ the following would hold:

$$p(\pi(\mathbf{e}_1|\omega_1)|\pi(\mathbf{e}_2|\omega_2)) = p(\pi(\mathbf{e}_1|\omega_1)|\pi(\mathbf{e}_2|\omega_3)). \tag{1}$$

$p(\pi(\mathbf{e}_1|\omega_1)|\pi(\mathbf{e}_2|\omega_2))$ can be understood as conditional probability that $\pi(\mathbf{e}_1|\omega_1)$ *is a representation of an image* $\mathbf{x}$ *transformed by* $t_{\omega_1}$, *given the knowledge that* $\pi(\mathbf{e}_2|\omega_2)$ *is a representation of* $\mathbf{x}$ *transformed by* $t_{\omega_2}$. Equation 1 implies that replacing the knowledge of $t_{\omega_2}$ with any other randomly sampled $t_{\omega_3}$ does not affect the inference process of CASSLE.

To demonstrate that this is not the case, we measure the similarity of representations of 5000 positive image pairs from the ImageNet-100 test set, where one representation is always constructed using true information about applied augmentations, and the second is constructed using either true or randomly sampled augmentation information. It is evident from Figure 3 that the similarity of embeddings decreases when false augmentation parameters ($\omega_3$ instead of $\omega_2$) are supplied to the projector, i.e:

$$
\begin{aligned}
\mathbb{E}_{x \sim \mathbb{X}, \{\omega_1, \omega_2, \omega_3\} \sim \Omega}\big[ \\
sim\left(\pi(\mathbf{e}_1|\omega_1), \pi(\mathbf{e}_2|\omega_2)\right) - \\
sim\left(\pi(\mathbf{e}_1|\omega_1), \pi(\mathbf{e}_2|\omega_3)\right) \\
\big] > 0.
\end{aligned}
\tag{2}
$$

Recall that in contrastive SSL, the similarity of embeddings corresponds to their probability density. This is because the InfoNCE loss is formulated as cross-entropy, where the activation function is defined as similarity between respective image embeddings, and class labels are replaced with the indices of corresponding positive embedding pairs (van den Oord et al., 2019; He et al., 2020; Chen et al., 2020a). Hence, minimizing $\mathcal{L}$ leads to:

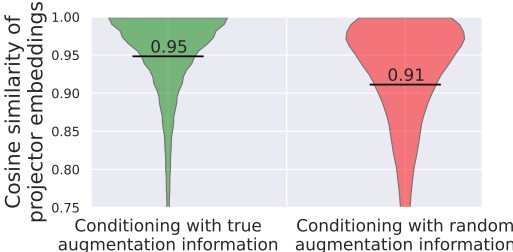

Figure 3: Similarities of CASSLE projector representations when conditioned with augmentation information from either their respective images (green), or randomly sampled (red). Solid lines denote the mean values of similarities. Guiding the CASSLE projector with wrong augmentation information decreases its ability to draw image pairs together, indicating that it indeed relies on augmentation information to perform its task.

$$sim\left(\pi(\mathbf{e}_1|\omega_1), \pi(\mathbf{e}_2|\omega_2)\right) \quad \propto \quad \frac{p(\pi(\mathbf{e}_1|\omega_1)|\pi(\mathbf{e}_2|\omega_2))}{p(\pi(\mathbf{e}_1|\omega_1))}. \tag{3}$$

It follows from 2 that, in practice

$$\mathbb{E}_{x \sim \mathbb{X}, \{\omega_1, \omega_2, \omega_3\} \sim \Omega}\left[\frac{p(\pi(\mathbf{e}_1|\omega_1)|\pi(\mathbf{e}_2|\omega_2)) - p(\pi(\mathbf{e}_1|\omega_1)|\pi(\mathbf{e}_2|\omega_3))}{p(\pi(\mathbf{e}_1|\omega_1))}\right] > 0 \tag{4}$$

and, since $p(\pi(\mathbf{e}_1|\omega_1)) > 0$,

$$\mathbb{E}_{x \sim \mathbb{X}, \{\omega_1, \omega_2, \omega_3\} \sim \Omega}\left[p(\pi(\mathbf{e}_1|\omega_1)|\pi(\mathbf{e}_2|\omega_2)) - p(\pi(\mathbf{e}_1|\omega_1)|\pi(\mathbf{e}_2|\omega_3))\right] > 0. \tag{5}$$

Moreover, we measure whether $p(\pi(\mathbf{e}_1|\omega_1)|\pi(\mathbf{e}_2|\omega_2)) > p(\pi(\mathbf{e}_1|\omega_1)|\pi(\mathbf{e}_2|\omega_3))$ for each of the considered image view pairs and find it to be true in 92% of the considered cases.

In CASSLE, *the conditional probability of matching a positive pair of image representations increases when the correct augmentation information is known*, which implies that information pertaining to augmented features is indeed preserved in the representation of it's feature extractor.

CASSLE can be applied to a variety of joint-embedding SSL methods, as the only practical modification it makes is changing the projector network to utilize the additional input $\omega$, describing the augmentations. We do not modify any other aspects of the self-supervised approaches, such as objective functions, which is appealing from a practical perspective. Last but not least, the architecture of the feature extractor in CASSLE is not affected by the introduced augmentation conditioning, as we only modify the input to the projector, which is discarded after the pretraining. Just like in vanilla SSL techniques, the feature extractor can be directly used in downstream tasks.

### 3.3 PRACTICAL IMPLEMENTATION OF THE CONDITIONING MECHANISM

In this work, we focus on a set of augmentations used commonly in the literature (Chen et al., 2020a;b; Chen & He, 2021), listed below along with descriptions of their respective parameters $\omega^{aug}$:

- **random cropping** – $\omega^c \in [0,1]^4$ describes the normalized coordinates of cropped image center and cropping sizes.
- **color jittering** – $\omega^j \in [0,1]^4$ describes the normalized intensities of brightness, contrast, saturation, and hue adjustment.
- **Gaussian blurring** – $\omega^b \in [0,1]$ is the standard deviation of the Gaussian filter used during the blurring operation.
- **random horizontal flipping** – $\omega^f \in \{0,1\}$ indicates whether the image has been flipped.
- **random grayscaling** – $\omega^g \in \{0,1\}$ indicates whether the image has been reduced to grayscale.

To enhance the projector's awareness of the color changes in the augmented images, we additionally enrich $\omega$ with information about **color difference** – $\omega^d \in [0,1]^3$, which is computed as the difference between the mean values of color channels of the image before and after the color jittering operation. We empirically demonstrate that inclusion of $\omega^d$ in $\omega$ improves the performance of CASSLE (see Section 4.3).

We construct augmentation information $\omega \in \Omega$ by concatenating vectors $\omega^{aug}$ describing the parameters of each augmentation type (Lee et al., 2021). We consider four methods of injecting $\omega$ into $\pi$: joining $\omega$ and $\mathbf{e}$ through **(i)** concatenation, modulating $\mathbf{e}$ with $\omega$ through element-wise **(ii)** addition or **(iii)** multiplication, or **(iv)** using $\omega$ as an input to a hypernetwork (Ha et al., 2017) which generates the parameters of $\pi$. Apart from concatenation, all of those methods require transforming $\omega$ into *augmentation embeddings* $\mathbf{g} \in \mathcal{G}$ of shapes required by the conditioning operation. For example, when modulating $\mathbf{e}$ with $\mathbf{g}$, dimensions of $\mathbf{e}$ and $\mathbf{g}$ must be equal. For this purpose, we precede the projector with additional *Augmentation encoder* $\gamma : \Omega \to \mathcal{G}$. For the architecture of $\gamma$ we choose the Multilayer Perceptron. An additional advantage of $\gamma$ is that it allows for learning a representation of $\omega$ which is more expressive for processing by $\pi$, which is typically a shallow network. In practice, we find that conditioning $\pi$ through the concatenation of $\mathbf{e}$ and $\mathbf{g}$ yields the best-performing representation (see Section 4.3).

## 4 EXPERIMENTS

In Section 4.1, we evaluate CASSLEs performance on downstream tasks such as classification, regression, and object detection[3]. In Section 4.2, we analyze the representations formed by CASSLE. Finally, we discuss the choice of hyperparameters of CASSLE in Section 4.3. In all experiments, unless specified otherwise, we utilize the ResNet-50 architecture (He et al., 2016) and conduct the self-supervised pretraining on ImageNet-100 - a 100-class subset of the ILSVRC dataset (Russakovsky et al., 2014) used commonly in the literature (Tian et al., 2020; Xiao et al., 2021; Lee et al., 2021; Chavhan et al., 2023). We use the standard set of augmentations including horizontal flipping, random cropping, grayscaling, color jittering and Gaussian blurring (He et al., 2020; Lee et al., 2021; Grill et al., 2020). For consistency in terms of hyperparameters, we follow (Lee et al., 2021) for MoCo-v2 , and (Chavhan et al., 2023) for SimCLR. We include comparisons on other SSL frameworks (He et al., 2020; Grill et al., 2020; Chen & He, 2021; Zbontar et al., 2021), extended analysis of trained representations, as well as additional details of training and evaluation, in the supplementary material.

### 4.1 EVALUATION ON DOWNSTREAM TASKS

We begin the experimental analysis by addressing the most fundamental question – how does CASSLE impact the ability of models to generalize? In order to answer it, we evaluate models

---

[3]We compare CASSLE to a number of recently proposed methods and report their performance from the literature (Xiao et al., 2021; Lee et al., 2021; Chavhan et al., 2023), given that the code for (Xiao et al., 2021) and (Chavhan et al., 2023) was not made available at the time of writing. As for the results of baseline SSL models and AugSelf (Lee et al., 2021), we report their results from the literature except when our runs of those methods yielded results different by at least 2 pp. We mark such cases with †.

Table 1: Linear evaluation on downstream classification and regression tasks. CASSLE consistently improves representations formed by vanilla SSL approaches and performs better or comparably to other techniques of increasing sensitivity to augmentations (Xiao et al., 2021; Lee et al., 2021; Chavhan et al., 2023).

| Method | C10 | C100 | Food | MIT | Pets | Flowers | Caltech | Cars | FGVCA | DTD | SUN | CUB | 300W |
|---|---|---|---|---|---|---|---|---|---|---|---|---|---|
| *SimCLR* (Chen et al., 2020a) | | | | | | | | | | | | | |
| Vanilla | 81.80 | 61.40 | 56.59[†] | 61.26[†] | 69.10 | 81.58[†] | 75.95[†] | 31.20[†] | 38.68[†] | 64.99[†] | 46.37[†] | 28.87[†] | 88.47[†] |
| AugSelf (Lee et al., 2021)[†] | 84.30 | 63.47 | 60.76 | 63.43 | **71.86** | **86.59** | 79.88 | 36.56 | **42.90** | 66.59 | 48.84 | 34.46 | 88.79 |
| AI (Chavhan et al., 2023) | 83.90 | 63.10 | – | – | 69.50 | 68.30 | 74.20 | – | – | 53.70 | – | **38.60** | 88.00 |
| **CASSLE** | **85.61** | **64.09** | **61.00** | **63.58** | 71.43 | 85.98 | **80.62** | 37.97 | 42.26 | **67.07** | **49.42** | 33.91 | **89.05** |
| *MoCo-v2* (He et al., 2020; Chen et al., 2020b) | | | | | | | | | | | | | |
| Vanilla | 84.60 | 61.60 | 59.67 | 61.64 | 70.08 | 82.43 | 77.25 | 33.86 | 41.21 | 64.47 | 46.50 | 32.20 | 88.77[†] |
| AugSelf (Lee et al., 2021) | 85.26 | 63.90 | 60.78 | 63.36 | 73.46 | 85.70 | 78.93 | 37.35 | 39.47 | 66.22 | 48.52 | 37.00 | 89.49[†] |
| AI (Chavhan et al., 2023) | 81.30 | 64.60 | – | – | **74.00** | 81.30 | 78.90 | – | – | **68.80** | – | **41.40** | **90.00** |
| LooC (Xiao et al., 2021) | – | – | – | – | – | – | – | – | – | – | 39.60 | – | – |
| IFM Robinson et al. (2021)[†] | 83.36 | 60.22 | 59.86 | 60.60 | 72.99 | 85.73 | 78.77 | 36.54 | 41.05 | 62.34 | 47.48 | 35.90 | 88.92 |
| **CASSLE** | **86.32** | **65.29** | **61.93** | **63.86** | 72.86 | **86.51** | 79.63 | **38.82** | 42.03 | 66.54 | **49.25** | 36.22 | 88.93 |
| *MoCo-v3* (Chen et al., 2021b) with ViT-Small (Dosovitskiy et al., 2021) pretrained on the full ImageNet dataset. | | | | | | | | | | | | | |
| Vanilla[†] | 83.17 | 62.40 | 56.15 | 53.28 | 62.29 | 81.48 | 69.63 | 28.63 | 32.84 | 57.18 | 42.16 | 35.00 | 87.42 |
| AugSelf (Lee et al., 2021)[†] | 84.25 | 64.12 | **58.28** | **56.12** | **63.93** | **83.13** | 72.45 | 29.64 | 32.54 | **60.27** | 43.22 | **37.16** | 87.85 |
| **CASSLE** | **85.13** | **64.67** | 57.30 | 55.90 | 63.88 | 82.42 | **73.53** | **30.92** | **35.91** | 58.24 | **43.37** | 36.09 | **88.53** |

pretrained via CASSLE and other self-supervised techniques on a variety of downstream visual tasks, such as classification, regression, and object detection.

**Linear evaluation** We evaluate the performance of pretrained networks on the downstream tasks of classification on a wide array of datasets: CIFAR10/100 (C10/100) (Krizhevsky, 2009), Food101 (Food) (Bossard et al., 2014), MIT67 (MIT) (Quattoni & Torralba, 2009), Oxford-IIIT Pets (Pets) (Parkhi et al., 2012), Oxford Flowers-102 (Flowers) (Nilsback & Zisserman, 2008), Caltech101 (Caltech) (Fei-Fei et al., 2006), Stanford Cars (Cars) (Krause et al., 2013), FGVC-Aircraft (FGVCA) (Maji et al., 2013), Describable Textures (DTD) (Cimpoi et al., 2014), SUN-397 (SUN) (Xiao et al., 2010), as well as regression on the 300 Faces In-the-Wild (300W) dataset (Sagonas et al., 2016). We follow the linear evaluation protocol (Kornblith et al., 2019; Chen et al., 2020a; Lee et al., 2021), described in detail in the supplementary material. We evaluate multiple self-supervised methods extended with CASSLE, as well as other recently proposed extensions which increase sensitivity to augmentations (Lee et al., 2021; Xiao et al., 2021; Chavhan et al., 2023) or prevent feature suppression in SSL Robinson et al. (2021). We report the full results in Tables 1 and 7. We find that in the vast majority of cases, CASSLE improves the performance of vanilla joint-embedding methods , as well as other other SSL extensions that foster augmentation sensitivity (Lee et al., 2021; Chavhan et al., 2023).

**Object detection** We next evaluate the pretrained networks on a more challenging task of object detection on the VOC 2007 dataset (Everingham et al.). We follow the training scheme of (He et al., 2020; Chen et al., 2020b), except that we only train the object detector modules and keep the feature extractor parameters fixed during training for detection to better compare the pretrained representations. We report the Average Precision (AP) (Lin et al., 2014) of models pretrained through MoCo-v2 and Sim-CLR (Chen et al., 2020a) with AugSelf (Lee et al., 2021) and CASSLE extensions in Table 2. The compared approaches yield similar results, with CASSLE representation slightly surpassing the vanilla methods and AugSelf.

Table 2: Average Precision of object detection on VOC dataset (Everingham et al.; Lin et al., 2014). CASSLE extension of MoCo-v2 and SimCLR outperforms the vanilla approaches and AugSelf extension by a small margin.

| Method | *MoCo-v2* | *SimCLR* |
|---|---|---|
| Vanilla | 45.12 | 44.78 |
| AugSelf (Lee et al., 2021) | 45.20 | 44.44 |
| **CASSLE** | **45.90** | **45.02** |

## 4.2 ANALYSIS OF REPRESENTATIONS FORMED BY CASSLE

We investigate the awareness of augmentation-induced data perturbations in the intermediate and final representations of pretrained networks. As a proxy metric for measuring this, we choose the

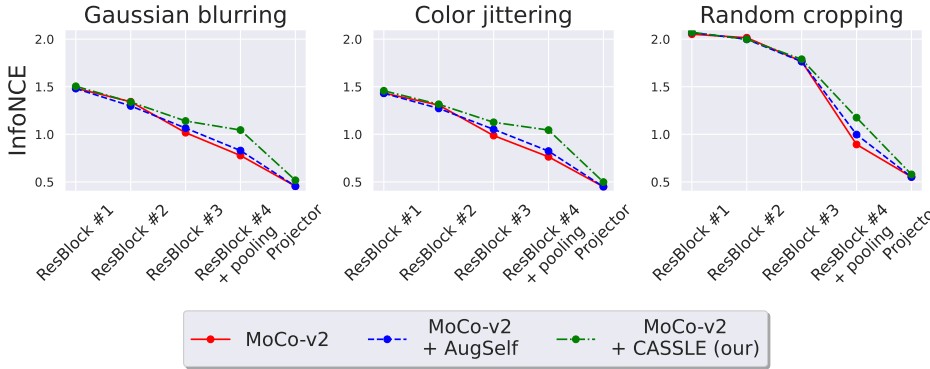

Figure 4: A comparison of InfoNCE loss measured on different kinds of augmentations at subsequent stages of the ResNet-50 and projectors pretrained by vanilla, AugSelf (Lee et al., 2021) and CASSLE variants of MoCo-v2. Feature extractor representation of CASSLE yields higher InfoNCE values which suggests that it is more susceptible to augmentations.

InfoNCE loss (van den Oord et al., 2019; Chen et al., 2020a). The value of InfoNCE is high if embeddings of pairs of augmented images are less similar to one another than to embeddings of unrelated images, and low if positive pairs of embeddings are on average separated correctly, and thus, the given representation is invariant to augmentations. We report the mean InfoNCE loss values under different augmentation types at subsequent stages of ResNet-50 and projectors of MoCo-v2, AugSelf (Lee et al., 2021) and CASSLE in Figure 4.

In all networks, the augmentation awareness decreases gradually throughout the feature extractor and projector stages. In CASSLE, we observe a much softer decline in the feature extractor stages and a sharper one in the projector. Representations of CASSLE feature extractor are on average more difficult to match together than those of vanilla MoCo-v2 and AugSelf (Lee et al., 2021). This implies that the CASSLE feature extractor is indeed more sensitive to augmentations than its counterparts. On the other hand, representations of all projectors, including CASSLE, are similarly separable. This suggests that the conditioning mechanism helps CASSLE projector to better amortize the augmentation-induced differences between the feature extractor embeddings.

The above observations indicate that in the vanilla and (to a slightly lesser extent) AugSelf approaches, both the projector and the intermediate representations are enforced to be augmentation-invariant. On the other hand, in CASSLE, the task of augmentation invariance is solved to a larger degree by the projector, and to a smaller degree by the feature extractor, allowing it to be more augmentation-aware. As shown in Section 4.1, this sensitivity does not prevent the CASSLE feature extractor from achieving similar or better performance than its counterparts when transferred to downstream tasks.

### 4.3 ABLATION STUDY

In this section, we examine the impact of different hyperparameters of CASSLE. We compare different variants of MoCo-v2+CASSLE on the same classification and regression tasks as in Section 4.1. We rank the models from best to worst performance on each task and report the average ranks in Table 3. We provide the detailed results in the supplementary material.

**Augmentation information contents** – We compare conditioning the projector with different subsets of augmentation information. The average best representation is trained with conditioning on all possible augmentation information. Moreover, using the additional **color difference** ($\omega^d$) information additionally improves the results, indicating that it is indeed useful to consider not only augmentation parameters but also information about its effects.

**Impact of utilizing color difference information** – We verify that the improved performance of CASSLE does not stem solely from using augmentation information that has not been considered in prior works. We compare a variant of AugSelf which learns to predict color difference values ($\omega^d$)

Table 3: Ablation study of CASSLE parameters. CASSLE performs best when conditioned on all available augmentation information, by concatenating or adding the augmentation and image embeddings.

| Augmentation information | Average rank ↓ |
|---|---|
| $\omega^{\{c\}}$ | 4.54 |
| $\omega^{\{c,j\}}$ | 4.54 |
| $\omega^{\{c,j,d\}}$ | 3.08 |
| $\omega^{\{c,j,b,f\}}$ | 2.85 |
| $\omega^{\{c,j,b,f,g\}}$ | 3.54 |
| $\omega^{\{c,j,b,f,g,d\}}$ | **2.54** |

(a) Augmentation information used for conditioning

| Method | Average rank ↓ |
|---|---|
| $A - \omega^{\{c,j\}}$ | 3.07 |
| $C - \omega^{\{c,j,b,f,g\}}$ | 2.64 |
| $A - \omega^{\{c,j,d\}}$ | 2.71 |
| $C - \omega^{\{c,j,b,f,g,d\}}$ | **1.57** |

(b) Impact of including color difference information ($\omega^d$) on **A**ugSelf and **CASSLE**

| Conditioning method | Average rank ↓ |
|---|---|
| Concatenation | **1.92** |
| Addition | **1.92** |
| Multiplication | 2.69 |
| Hypernetwork | 3.46 |

(c) Method of conditioning the projector

| Number of layers | Average rank ↓ |
|---|---|
| 0 | 3.38 |
| 2 | 3.08 |
| 4 | 2.38 |
| 6 | **2.15** |
| 8 | 4.00 |

(d) Augmentation encoder depth (0 denotes concatenating raw $\omega$ to **e**)

in addition to augmentation information used by (Lee et al., 2021), i.e. cropping ($\omega^c$) and color jittering ($\omega^j$), as well as a variant of CASSLE conditioned on all augmentation information *except* $\omega^d$ (i.e. $\omega^{\{c,j,b,f,g\}}$). We find that while including $\omega^d$ improves the performance of AugSelf and CASSLE, both variants of CASSLE achieve better results than both variants of AugSelf.

**Method of conditioning the projector** – We compare conditioning the projector through (i) concatenation, element-wise (ii) addition or (iii) multiplication, or (iv) using $\pi$ as an input to a hypernetwork (Ha et al., 2017) which generates the parameters of $\pi$. Conditioning through **concatenation** and **addition** yields on average the strongest performance on downstream tasks. We choose to utilize the **concatenation** method in our experiments, as it requires a slightly smaller Augmentation encoder.

**Size of the Augmentation encoder** – While CASSLE is robust to the size of the $\gamma$ MLP, using the depth and hidden size of 6 and 64, respectively, yields the strongest downstream performance. In particular, the variant of CASSLE that utilizes the Augmentation encoder performs better than the variant that concatenates **e** to raw augmentation embeddings $\omega$. Given such an architecture of the Augmentation encoder, our computation overhead is negligible as we increase the overall number of parameters by around $0.1\%$. We refer to the supplementary material for the detailed ablation of the Augmentation encoder's hidden size.

## 5 CONCLUSION

In this paper, we propose a novel method for augmentation-aware self-supervised learning that retains information about data augmentations in the representation space. To accomplish this, we introduce the concept of the conditioned projector, which receives augmentation information while processing the representation vector. Our solution necessitates only small architectural changes and no additional auxiliary loss components. Therefore, the training concentrates on contrastive loss, which enhances overall performance.

We compare our solution with existing augmentation-aware SSL methods and demonstrate its superior performance on downstream tasks, particularly when augmentation invariance leads to the loss of vital information. Moreover, we show that it obtains representations more sensitive to augmentations than the baseline methods.

Overall, our method offers a straightforward and efficient approach to retaining information about data augmentations in the representation space. It can be directly applied to SSL methods, contributing to the further advancement of augmentation-aware self-supervised learning.

**Reproducibility statement** The anonymized codebase for CASSLE is available at `https://anonymous.4open.science/r/CASSLE-037C`. We detail the hyperparameters for training runs in the supplementary material. All datasets used for training and evaluation are publicly available.

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

# A  PRETRAINING DETAILS

## A.1  DATASETS

We use ImageNet-100, a 100-class subset of ImageNet (Russakovsky et al., 2014; Tian et al., 2020), to pretrain the standard ResNet-50 (He et al., 2016) architecture of self-supervised methods: MoCo-v2 (Chen et al., 2020b), SimCLR (Chen et al., 2020a), Barlow Twins (Zbontar et al., 2021), and SimSiam (Chen & He, 2021), as well as for common in the literature on augmentation-aware self-supervised learning (Tian et al., 2020; Xiao et al., 2021; Lee et al., 2021; Chavhan et al., 2023). For MoCo-v3 (Chen et al., 2021b), we pretrain the ViT-Small (Dosovitskiy et al., 2021) model on the full ImageNet dataset (Russakovsky et al., 2014).

## A.2  HYPERPARAMETERS

We follow the pretraining procedures from corresponding papers, described in (Lee et al., 2021) for MoCo-v2 and SimSiam, (Chavhan et al., 2023) for SimCLR, (Zbontar et al., 2021) for Barlow Twins, and (Chen et al., 2021b) for MoCo-v3. CASSLE is trained with ResNet50 and ViT-small architectures on 2 and 8 NVidia A100 GPUs, respectively. Synchronized batch normalization is employed for distributed training (Chen & He, 2021). In Table 4, we present the training hyperparameters which are not related specifically to CASSLE, but rather joint-embedding approaches in general (He et al., 2020; Chen et al., 2020b;a; Zbontar et al., 2021; Chen & He, 2021; Chen et al., 2021b).

Table 4: Hyperparameters of self-supervised methods used with CASSLE

| SSL method | Architecture | Number of epochs | Batch size | Weight decay | Learning rate | | Training time |
|---|---|---|---|---|---|---|---|
| | | | | | *Base* | *Schedule* | |
| MoCo-v2 (Chen et al., 2020b) | ResNet-50 | 500 | 256 | $10^{-4}$ | 0.03 | Cosine decay | 34h |
| SimCLR (Chen et al., 2020a) | ResNet-50 | 300 | 256 | $10^{-4}$ | 0.05 | Cosine decay | 48h |
| Barlow Twins (Zbontar et al., 2021) | ResNet-50 | 500 | 256 | $10^{-4}$ | 0.05 | Cosine decay with warmup | 36h |
| SimSiam (Chen & He, 2021) | ResNet-50 | 500 | 256 | $10^{-4}$ | 0.05 | Cosine decay | 40h |
| MoCo-v3 (Chen et al., 2021b) | ViT-small | 300 | 1024 | 0.1 | $1.5 \cdot 10^{-4}$ | Cosine decay with warmup | 23h |

| SSL method | Projector | | | | Predictor | | | |
|---|---|---|---|---|---|---|---|---|
| | *Depth* | *Hidden size* | *Out size* | *Final BatchNorm* | *Depth* | *Hidden size* | *Out size* | *Final BatchNorm* |
| MoCo-v2 | 2 | 2048 | 128 | No | | | | No |
| SimCLR | 2 | 2048 | 128 | No | | | | No |
| Barlow Twins | 3 | 8192 | 8192 | Yes, without affine transform | | | | No |
| SimSiam | 3 | 2048 | 2048 | Yes | 2 | 512 | 2048 | No |
| MoCo-v3 | 3 | 4096 | 256 | Yes, without affine transform | 2 | 4096 | 256 | Yes, without affine transform |

## A.3  AUGMENTATIONS

For self-supervised pretraining, we use a set of augmentations adopted commonly in the literature (He et al., 2020; Chen et al., 2020a;b; Chen & He, 2021; Zbontar et al., 2021; Lee et al., 2021). We denote them below:

- **random cropping** – We sample the cropping scale randomly from [0.2, 1.0]. Afterward, we resize the cropped images to the size of $224 \times 224$.

- **color jittering** – We apply this operation with a probability of $0.8$. We sample the intensities of brightness, contrast, saturation, and hue and their maximal values are $0.4$, $0.4$, $0.4$, and $0.1$, respectively.

- **Gaussian blurring** – We apply this operation with a probability of $0.5$. We sample the standard deviation from $[0.1, 2.0]$ and set the kernel size to $23 \times 23$.

- **random horizontal flipping** – We apply this operation with a probability of $0.5$.

- **random grayscaling** – We apply this operation with a probability of $0.2$.

## B    Evaluation protocol

### B.1    Datasets

We outline the datasets and their respective evaluation metrics used for downstream tasks that we evaluate CASSLE on in Table 5.

Table 5: Information about datasets and their metrics used for downstream evaluation of CASSLE.

| Downstream task | Dataset | Metric |
|---|---|---|
| Linear evaluation | CIFAR10 (Krizhevsky, 2009) | Top-1 accuracy |
| | CIFAR100 (Krizhevsky, 2009) | Top-1 accuracy |
| | Food101 (Bossard et al., 2014) | Top-1 accuracy |
| | MIT67 (Quattoni & Torralba, 2009) | Top-1 accuracy |
| | Oxford-IIIT Pets (Parkhi et al., 2012) | Mean per-class accuracy |
| | Oxford Flowers-102 (Nilsback & Zisserman, 2008) | Mean per-class accuracy |
| | Caltech101 (Fei-Fei et al., 2006) | Mean per-class accuracy |
| | Stanford Cars (Krause et al., 2013) | Top-1 accuracy |
| | FGVC-Aircraft (Maji et al., 2013) | Mean Per-class accuracy |
| | Describable Textures (split 1) (Cimpoi et al., 2014) | Top-1 accuracy |
| | SUN397 (split 1) (Xiao et al., 2010) | Top-1 accuracy |
| | Caltech-UCSD Birds (Wah et al., 2011) | Top-1 accuracy |
| | 300 Faces In-the-Wild (Sagonas et al., 2016) | $R^2$ |
| Object detection | VOC2007 (Everingham et al.) | Average Precision (Lin et al., 2014) |
| Few-shot classification | Few-Shot CIFAR100 (Oreshkin et al., 2018) | Average accuracy |
| | Caltech-UCSD Birds (Wah et al., 2011) | Average accuracy |
| | Plant Disease (Mohanty et al., 2016) | Average accuracy |
| | Oxford Flowers-102 (Nilsback & Zisserman, 2008) | Mean per-class accuracy |

**Linear evaluation**    We follow the linear evaluation protocol of (Chen et al., 2020a; Grill et al., 2020; Kornblith et al., 2019; Lee et al., 2021). Namely, we center-crop and resize the images from the downstream dataset to the size of $224 \times 224$, pass them through the pretrained feature extractor, and obtain the embeddings from the final feature extractor stage. The only exception from this is the CUB dataset (Wah et al., 2011), where, following (Lee et al., 2021), for the training images besides the center crop of the image, we also crop the image at its corners and do the same for the horizontal flip of the image (this is known as TenCrop operation[4]).

Having gathered the image features, we minimize the $l_2$-regularized cross-entropy objective using L-BFGS on the features of the training images. We select the regularization parameter from between $[10^{-6}, 10^5]$ using the validation features. Finally, we train the linear classifier on training and validation features with the selected $l_2$ parameter and report the final performance metric (see Table 5) on the test dataset. We set the maximum number of iterations in L-BFGS as 5000 and use the model trained on training data as initialization for training the final model.

We note that the above linear evaluation procedure is effectively equivalent to training the final layer of a network on non-augmented data, while keeping the remainder of the parameters unchanged.

**Object detection**    We closely follow the evaluation protocol of (He et al., 2020). We train the Faster-RCNN (Ren et al., 2015) model with the pretrained backbone. Contrary to (He et al., 2020), we do not tune the backbone parameters, in order to better observe the effect of different pretraining methods. We report the Average Precision (Lin et al., 2014) measured on the VOC `test2007` set (Everingham et al.).

**Sensitivity to augmentations**    We consider image pairs, where one image is the (center-cropped) original and the second one is augmented by the given augmentation. For each image pair, we extract image features at four stages of the pretrained ResNet-50 backbone (He et al., 2016), as well as the final representation of the projector network. We next calculate cosine similarities between the

---

[4]`https://pytorch.org/vision/main/generated/torchvision.transforms.TenCrop.html`

features of augmented and non-augmented images in the given mini-batch (of size 256). We report the value of the InfoNCE loss (van den Oord et al., 2019) calculated on such similarity matrices.

**Dependency of CASSLE projector on conditioning**   Similarly to the above experiment, we compare the projector features of augmented and non-augmented image pairs. When computing the features of the augmented image, we supply the projector with the augmentation embedding computed from augmentation parameters corresponding to either this image (true augmentation information) or another, randomly chosen image from the same mini-batch (random augmentation information). We then compute cosine similarities between the original image features and features of the augmented image computed with true/random augmentation information.

## C   ADDITIONAL RESULTS AND ANALYSIS

### C.1   EVALUATION

**Rotation prediction**   In order to understand whether CASSLE learns to generalize to other types of augmentation, which were not used during pretraining, we inspect the prediction of applied augmentation through the following experiment. We train a linear classifier on top of the pretrained model to indicate whether the image was rotated by 0, 90, 180, or 270 degrees. We formulate the problem as classification due to its cyclic nature and test the model on the datasets used for linear evaluation in Section 4.1 of the main manuscript. We present the results of vanilla, AugSelf (Lee et al., 2021) and CASSLE variants of self-supervised methods in Table 6. Apart from a few exceptions, CASSLE and AugSelf extensions allow in general for better prediction of rotation than the vanilla SSL methods. Moreover, in the case of SimCLR, CASSLE representation predicts the rotations the most accurately on a vast majority of datasets. This occurs despite the fact, that neither of the methods was trained using rotated images and thus, never explicitly learned the *concept* of rotation. This suggests that CASSLE learns representations which are sensitive to a broader set of perturbations than those whose information had been used in CASSLE's pretraining.

Table 6: Linear evaluation on downstream task: predicting the rotation (0, 90, 180 or 270 degrees)

| Method | C10 | C100 | Food | Pets | MIT | Flowers | Caltech | Cars | FGVCA | STL10 | SUN |
|---|---|---|---|---|---|---|---|---|---|---|---|
| | | | | | *SimCLR* (Chen et al., 2020a) | | | | | | |
| Vanilla | 67.93 | 56.88 | 68.76 | 65.07 | **70.40** | 24.15 | 56.81 | 87.12 | 98.02 | **73.90** | 67.94 |
| AugSelf (Lee et al., 2021) | 75.63 | 63.08 | 75.01 | 57.99 | 16.54 | **33.99** | 39.03 | 76.11 | 96.64 | 67.88 | 76.60 |
| **CASSLE** | **77.07** | **70.20** | **79.55** | **70.97** | 68.77 | 33.08 | **64.97** | **95.14** | **99.46** | 72.69 | **78.16** |
| | | | | | *MoCo-v2* (He et al., 2020; Chen et al., 2020b) | | | | | | |
| Vanilla | 67.96 | 56.96 | 75.70 | 71.27 | 67.13 | **58.30** | **87.33** | 58.35 | 93.20 | **95.44** | 68.33 |
| AugSelf (Lee et al., 2021) | **74.57** | 65.87 | 73.03 | **77.01** | **79.75** | 52.81 | 58.53 | **93.07** | 98.26 | 77.35 | **83.66** |
| **CASSLE** | 73.21 | **69.91** | **77.31** | 76.04 | 76.40 | 45.11 | 61.06 | 90.49 | **98.56** | 63.65 | 76.01 |
| | | | | | *SimSiam* (Chen & He, 2021) | | | | | | |
| Vanilla | 68.11 | 64.81 | 77.35 | 61.79 | **79.10** | 46.33 | **64.76** | 80.50 | 97.48 | 55.38 | 74.49 |
| AugSelf (Lee et al., 2021) | **77.00** | 72.91 | 73.03 | 59.18 | 78.28 | **49.11** | 60.86 | **97.21** | **99.46** | **76.68** | **83.14** |
| **CASSLE** | 72.00 | **71.38** | **78.01** | **85.30** | 66.80 | 37.65 | 61.64 | 92.04 | 98.65 | 63.69 | 75.86 |
| | | | | | *Barlow Twins* (Zbontar et al., 2021) | | | | | | |
| Vanilla | 73.67 | 64.87 | 72.85 | **83.02** | 70.97 | 42.56 | **57.71** | 83.68 | 97.93 | 65.70 | 76.87 |
| AugSelf (Lee et al., 2021) | 72.79 | **65.91** | 74.93 | 77.76 | 50.30 | 30.93 | 44.31 | 85.85 | 98.35 | **69.68** | **77.19** |
| **CASSLE** | **74.97** | 65.15 | **76.55** | 77.05 | **86.49** | **43.16** | 53.60 | **92.92** | **99.16** | 55.24 | 74.97 |

**Linear evaluation – additional SSL frameworks**   We evaluate CASSLE on additional joint-embedding SSL frameworks (He et al., 2020; Grill et al., 2020; Chen & He, 2021; Zbontar et al., 2021), following the same linear evaluation protocol as in Section 4.1, and present the results in Table 7. In all cases, CASSLE improves the performance of the respective Vanilla approaches. In the case of MoCo-v1 and Barlow Twins, it also achieves stronger performance than AugSelf (Lee et al., 2021), whereas in the case of self-distillation methods (Grill et al., 2020; Chen & He, 2021), the performances are comparable. We suspect that the lower performance of CASSLE compared

to AugSelf with those methods stems from their asymmetric architecture, i.e., distilling the representations between the projector and an additional predictor network. Observe that CASSLE and AugSelf also perform comparably on MoCo-v3, which also utilizes an additional predictor (see Table 1). Perhaps most curiously, CASSLE lends a major performance boost to MoCo-v1 (He et al., 2020). This is surprising, as the original MoCo-v1 architecture does not utilize a projector network. As such, in this particular case, CASSLE joins the image and augmentation embeddings through element-wise addition. Nevertheless, this simple modulation improves the results by a large margin.

We summarize the linear evaluation results in Figure 5[5]. CASSLE is usually ranked the best in terms of performance on downstream tasks compared to AugSelf Lee et al. (2021) and Vanilla approaches. AugSelf and CASSLE improve over Vanilla approaches by a comparable margin. Finally, CASSLE achieves the best performance in the largest number of downstream tasks.

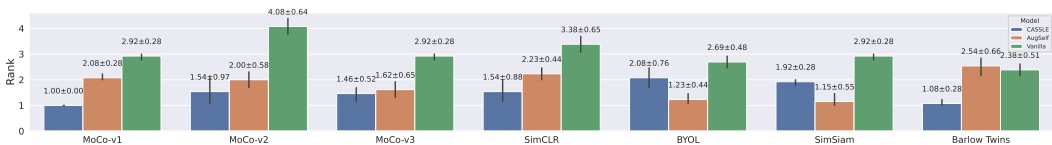

(a) Average ranks (with 95% confidence intervals) of CASSLE, AugSelf, and Vanilla models when ranked from best to worst on respective downstream tasks. Lower is better.

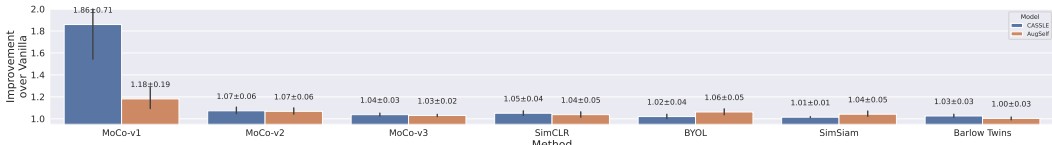

(b) Average relative performance (with 95% confidence intervals) to Vanilla models of CASSLE and AugSelf models.

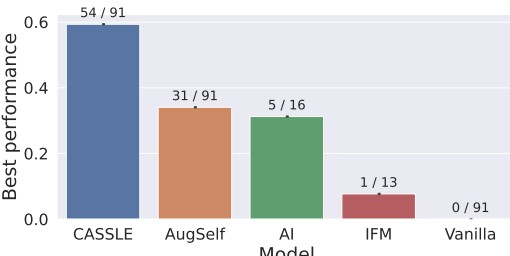

(c) Number of times (out of the times when the model is used in comparison) each of the models achieves the best performance (compared to other models) on downstream tasks. Higher is better.

Figure 5: Summary of linear evaluation of different SSL extensions. We compare the average performance ranks of extensions on the respective downstream tasks (Fig. 5a), average relative performance to Vanilla approaches (Fig. 5b), and the number of times each of the extensions achieves the best performance on a downstream task (Fig. 5c).

**Few-shot classification**  We next benchmark various forms of pretraining on a number of few-shot learning tasks: Few-Shot CIFAR100 (FC100) (Oreshkin et al., 2018), Caltech-UCSD Birds (CUB) (Wah et al., 2011), Plant Disease (Mohanty et al., 2016). For the above datasets, we report the performance on 5-way 1-shot and 5-way 5-shot tasks. Moreover, we also report the performance on the 5-shot and 10-shot versions of the Flowers-102 dataset (Flowers) (Nilsback & Zisserman, 2008). Following (Lee et al., 2021), our few-shot models are trained through logistic regression on representations of support images, without fine-tuning. We follow the evaluation protocol of (Lee et al., 2021). For the task of 5/10-shot learning of the Flowers dataset (Nilsback & Zisserman, 2008), we select the subset of the training set with 5/10 examples per class, respectively, and utilize the linear evaluation protocol presented above. For the remaining datasets used for few-shot evaluation,

---

[5]We include AI Chavhan et al. (2023) and IFM Robinson et al. (2021) only in Figure 5c, due to lack of their results for some of the SSL frameworks.

Table 7: Linear evaluation on downstream classification and regression tasks. CASSLE consistently improves representations formed by vanilla SSL approaches and performs better or comparably to other techniques of increasing sensitivity to augmentations (Xiao et al., 2021; Lee et al., 2021; Chavhan et al., 2023).

| Method | C10 | C100 | Food | MIT | Pets | Flowers | Caltech | Cars | FGVCA | DTD | SUN | CUB | 300W |
|---|---|---|---|---|---|---|---|---|---|---|---|---|---|
| *MoCo-v1* (He et al., 2020) (no projector, **e** and $\omega$ joined through addition) | | | | | | | | | | | | | |
| Vanilla[†] | 58.82 | 28.09 | 25.90 | 31.04 | 47.25 | 33.29 | 44.41 | 5.00 | 10.98 | 36.86 | 19.00 | 9.16 | 88.05 |
| AugSelf | 64.94 | 37.01 | 32.84 | 33.13 | 45.95 | 38.59 | 45.15 | 8.33 | 15.14 | 40.37 | 20.48 | 11.27 | 88.12 |
| **CASSLE** | **80.53** | **53.55** | **52.11** | **51.94** | **57.58** | **60.56** | **60.33** | **18.68** | **28.68** | **53.94** | **36.71** | **18.88** | **88.21** |
| *BYOL* (Grill et al., 2020) | | | | | | | | | | | | | |
| Vanilla | 83.48 | 60.17 | 57.79 | 62.84 | 75.70 | 82.47 | 78.80 | 38.38 | 38.16 | 63.46 | 46.97 | 34.12 | 88.81 |
| AugSelf (Lee et al., 2021) | **85.55** | **65.03** | **62.59** | **64.33** | **77.48** | **87.48** | 81.13 | **41.06** | **43.39** | 64.63 | **49.31** | **40.90** | 89.28 |
| **CASSLE** | 84.20 | 62.00 | 57.35 | 60.97 | 73.29 | 85.44 | **81.52** | 40.18 | 42.11 | **65.59** | 47.63 | 33.36 | **93.03** |
| *SimSiam* (Chen & He, 2021) | | | | | | | | | | | | | |
| Vanilla | 86.89 | 66.33 | 61.48 | 65.75 | 74.69 | 88.06 | 84.13 | 48.20 | 48.63 | 65.11 | 50.60 | 38.40 | 89.01 |
| AugSelf (Lee et al., 2021) | **88.80** | **70.27** | **65.63** | **67.76** | **76.34** | **90.70** | **85.30** | 47.52 | **49.76** | **67.29** | **52.28** | **45.30** | 92.84 |
| **CASSLE** | 87.38 | 67.36 | 63.27 | 66.84 | 75.02 | 88.95 | 84.86 | **48.51** | 49.35 | 66.81 | 51.62 | 39.47 | 89.37 |
| *Barlow Twins* (Zbontar et al., 2021) | | | | | | | | | | | | | |
| Vanilla[†] | 85.90 | 66.10 | 59.41 | 61.72 | 72.30 | 87.13 | 81.95 | 41.54 | 44.40 | 65.85 | 49.18 | 35.02 | 89.04 |
| AugSelf (Lee et al., 2021)[†] | **87.28** | 66.98 | 60.52 | 63.96 | 72.11 | 86.68 | 81.73 | 39.88 | 44.23 | 65.21 | 47.71 | 37.02 | 88.88 |
| **CASSLE** | 87.03 | **67.27** | **62.19** | **65.08** | **72.75** | **87.99** | **82.56** | **41.68** | **46.63** | **66.31** | **50.09** | **38.25** | **89.52** |

we extract the representations of support and query images with the feature extractor and train a logistic regression model[6], which we evaluate on the query set. We report the average accuracy from 2000 $N$-way $K$-shot tasks (Song et al., 2023) in Table 8. In general, CASSLE improves the performance of the vanilla joint-embedding approaches, but to a lesser extent than AugSelf (Lee et al., 2021) or AI (Chavhan et al., 2023). The notable exception is Barlow Twins (Zbontar et al., 2021), where CASSLE yields the strongest performance boost.

**Fine-tuning** In addition to linear evaluation, we also evaluate the models pretrained with MoCo-v2 under the L2SP fine-tuning paradigm (LI et al., 2018) on Caltech-101 (Fei-Fei et al., 2006), CIFAR-10 (Krizhevsky, 2009), CUB-200 (Wah et al., 2011), Food-101 (Bossard et al., 2014), MIT-67 (Quattoni & Torralba, 2009), and Pets (Parkhi et al., 2012) datasets. We follow the hyperparameters of Li. et. al. (LI et al., 2018), i.e. $\beta = 0.01$, $\alpha \in \{0.001, 0.01, 0.1, 1\}$, and learning rate $\in \{0.005, 0.01, 0.02\}$. We report our results in Table 9. We find that while fine-tuned networks differ in performance by a lesser margin than during linear evaluation, CASSLE-pretrained networks perform favorably in 8 out of 18 cases, more than either vanilla MoCo-v2 or AugSelf.

**Robustness under perturbations** We next verify the influence of increased sensitivity to augmentations on the robustness of MoCo-v2-pretrained models to perturbations. Following the experimental setup of (Lee et al., 2021), we train the pretrained networks for classification of ImageNet-100 and evaluate them on weather-corrupted images (fog, frost, snow, and brightness) (Hendrycks & Dietterich, 2019) from the validation set. We report the results in Table 10. We find that the network pretrained with CASSLE achieves the best results when dealing with images perturbed by brightness and snow, whereas vanilla MoCo-v2 performs best on images perturbed by fog and frost.

**Image retrieval** Finally, we evaluate the pretrained models on the task of image retrieval. We gather the features of images from the Cars and Flowers test sets and for a given query image, select four images closest to it in terms of the cosine distance of final feature extractor representations. We compare the images retrieved by MoCo-v2, AugSelf (Lee et al., 2021) and CASSLE in Figure 6. CASSLE selects pictures of cars that are the most consistent in terms of color. In the case of flowers, the nearest neighbor retrieved by the vanilla model is a different species than that of the first query image, whereas both CASSLE and AugSelf select the first two nearest neighbors from the same species but then retrieve images of flowers with similar shapes, but different colors. This again indicates greater reliability of features learned by CASSLE. For subsequent queries, CASSLE and AugSelf retrieve in general more consistently looking images, in particular in terms of color

---

[6]We use the Logistic Regression implementation from scikit-learn (Pedregosa et al., 2011).

Table 8: Few-shot classification accuracy averaged over 2000 tasks on Few-Shot CIFAR100 (FC100) (Oreshkin et al., 2018), Caltech-UCSD Birds (CUB) (Wah et al., 2011), Plant Disease (Mohanty et al., 2016), and accuracy on 5-shot and 10-shot versions of Flowers-102 dataset (Flowers) (Nilsback & Zisserman, 2008). (N, K) denotes N-way K-shot tasks. The best performance in each group is **bolded**.

| Method | FC100 | | CUB | | Plant Disease | | Flowers | |
|---|---|---|---|---|---|---|---|---|
| | (5,1) | (5,5) | (5,1) | (5,5) | (5,1) | (5,5) | (102, 5) | (102,10) |
| *MoCo v2* (He et al., 2020; Chen et al., 2020b) | | | | | | | | |
| Vanilla | 31.67 | 43.88 | 41.67 | 56.92 | 65.73 | 84.98 | 66.83$^\dagger$ | 78.80$^\dagger$ |
| AugSelf (Lee et al., 2021) | 35.02 | 48.77 | 44.17 | 57.35 | 71.80 | 87.81 | **72.95**$^\dagger$ | **82.12**$^\dagger$ |
| AI (Chavhan et al., 2023) | **37.40** | 48.40 | **45.00** | **58.00** | **72.60** | **89.10** | – | – |
| LooC (Xiao et al., 2021) | – | – | – | – | – | – | 70.90 | 80.80 |
| **CASSLE** | 35.15 | **49.30** | 42.98 | 55.82 | 70.74 | 87.12 | 72.36 | 82.07 |
| *SimCLR* (Chen et al., 2020a) | | | | | | | | |
| Vanilla $^\dagger$ | 32.47 | 44.46 | 40.83 | 53.58 | 70.82 | 87.53 | 70.73 | 80.72 |
| AugSelf (Lee et al., 2021)$^\dagger$ | **36.72** | **49.15** | **42.80** | **57.08** | 73.85 | **89.46** | **73.86** | 82.91 |
| **CASSLE** | 34.24 | 47.34 | 41.26 | 55.98 | **73.89** | 89.20 | 73.23 | **83.34** |
| *Barlow Twins* (Zbontar et al., 2021) | | | | | | | | |
| Vanilla $^\dagger$ | 38.08 | 53.93 | 43.20 | 61.95 | 70.98 | 91.06 | 75.28 | 83.43 |
| AugSelf (Lee et al., 2021)$^\dagger$ | 37.48 | **54.16** | 42.50 | 61.86 | 71.28 | 91.16 | 74.49 | 83.62 |
| **CASSLE** | **38.58** | 54.03 | **43.22** | **62.65** | **73.47** | **92.17** | **76.62** | **85.50** |
| *SimSiam* (Chen & He, 2021) | | | | | | | | |
| Vanilla | 36.19 | 50.36 | 45.56 | 62.48 | 75.72 | 89.94 | 75.53$^\dagger$ | 85.90 |
| AugSelf (Lee et al., 2021) | **39.37** | **55.27** | **48.08** | **66.27** | **77.93** | **91.52** | **79.67**$^\dagger$ | **88.30** |
| **CASSLE** | 36.18 | 50.16 | 44.93 | 60.07 | 76.43 | 90.82 | 77.96 | 86.64 |

Table 9: Results of fine-tuning different variants of MoCo-v2 with L2SP protocol (LI et al., 2018).

| Method | Caltech-101 | CIFAR-10 | CUB-200 | Food-101 | MIT-67 | Pets |
|---|---|---|---|---|---|---|
| MoCo-v2 | | | | | | |
| **Vanilla** | **89.80** | 94.65 | 56.31 | 74.36 | 61.79 | 82.97 |
| **AugSelf** | 89.16 | 94.36 | **56.67** | **74.94** | **63.73** | **83.27** |
| **CASSLE** | 89.27 | **94.73** | 55.93 | 74.07 | 61.19 | 82.86 |
| Barlow Twins | | | | | | |
| **Vanilla** | **86.54** | 91.67 | 48.02 | 55.68 | **63.43** | 77.98 |
| **AugSelf** | 85.82 | 91.94 | 48.23 | 56.35 | 62.91 | 77.84 |
| **CASSLE** | 86.01 | **92.03** | **54.15** | **59.83** | 62.54 | **78.71** |
| SimCLR | | | | | | |
| **Vanilla** | **90.67** | 94.75 | 54.55 | 75.36 | 63.51 | 83.65 |
| **AugSelf** | 90.38 | 94.76 | **56.36** | 75.86 | 62.91 | **84.25** |
| **CASSLE** | 90.47 | **95.19** | 55.65 | **76.78** | **63.88** | 83.51 |

scheme. However, for some distinct images, such as the last presented image of flowers, all methods have retrieved the correct results for the given query.

## C.2 EXTENDED ANALYSIS OF REPRESENTATIONS FORMED BY CASSLE

**Analysis of dimensional collapse of representations** Augmentation-invariant SSL models are known to suffer from dimensional collapse of their representations, due to the usage of strong augmentations and implicit regularization (Jing et al., 2022). Here, we analyze the effect of CASSLE on

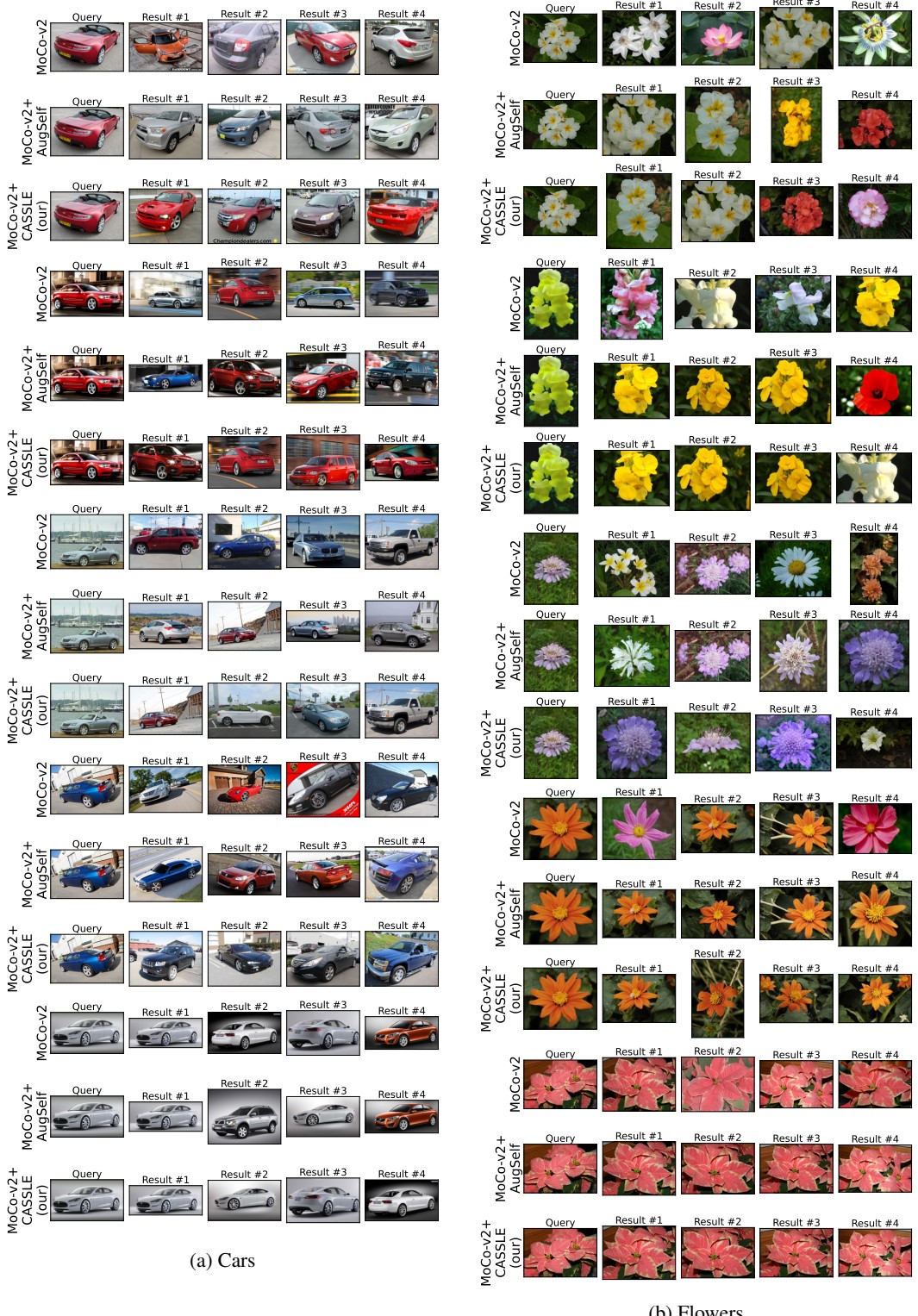

(a) Cars

(b) Flowers

Figure 6: Image retrieval examples for Cars and Flowers datasets.

Table 10: Evaluation of variants of MoCo-v2 on perturbed ImageNet-100 images.

| Method | Brightness | Frost | Fog | Snow |
|--------|------------|-------|-----|------|
| **Vanilla** | 85.30 | **53.70** | **56.92** | 31.78 |
| **AugSelf** | 83.64 | 51.98 | 53.08 | 33.80 |
| **CASSLE** | **86.10** | 50.54 | 54.22 | **34.66** |

this phenomenon. To this end, we compute the embeddings of the ImageNet-100 test dataset with pretrained feature extractors and calculate the empirical eigenspectra of the embeddings produced by each encoder. We next analyze the power-law decay rates ($\alpha$) of eigenspectra (Agrawal et al., 2022). Intuitively, smaller values of $\alpha$ suggest a dense encoding, while larger values of $\alpha$ suggest more sparse encoding and rapid decay. We report the calculated coefficients in Table 11. In all analyzed SSL methods, except Barlow Twins, the addition of augmentation information by CASSLE leads to a representation whose eigenspectrum decays more rapidly, thus suggesting a lower rank of the effective embedding subspace and as such, greater dimensional collapse. However, as seen in the main text, this does not necessarily lead to degraded performance – indeed, performance on downstream tasks has been previously correlated with how close $\alpha$ is to 1 (Agrawal et al., 2022).

Table 11: Power-law eigenspectrum decay rates ($\alpha$) (Agrawal et al., 2022) of embeddings of ImageNet-100 test set produced by differently pretrained feature extractors.

| Method | MoCo-v2 (He et al., 2020; Chen et al., 2020b) | SimCLR (Chen et al., 2020a) | Barlow Twins (Zbontar et al., 2021) | SimSiam (Chen & He, 2021) |
|--------|--------|--------|--------|--------|
| Vanilla | 0.6013 | 0.6150 | **0.5387** | 0.6365 |
| AugSelf (Lee et al., 2021) | 0.6146 | 0.6251 | 0.5201 | 0.6262 |
| **CASSLE** | **0.6342** | **0.6908** | 0.5196 | **0.6962** |

We illustrate this further with a Principal Component Analysis of features extracted by different variants of joint-embedding methods illustrated in Figure 7. We extract embeddings of the test ImageNet-100 images and calculate the ratio of variance explained by principal components of embeddings. We find that, in general, representations learned with CASSLE require fewer principal components to explain larger amounts of variance. For example, for MoCo-v2, 90% of variance is explained by 905, 872, and 736 principal components of representations learned by the vanilla approach, AugSelf (Lee et al., 2021), and CASSLE, respectively. This again suggests that representations learned by CASSLE have lower effective rank compared to other approaches.

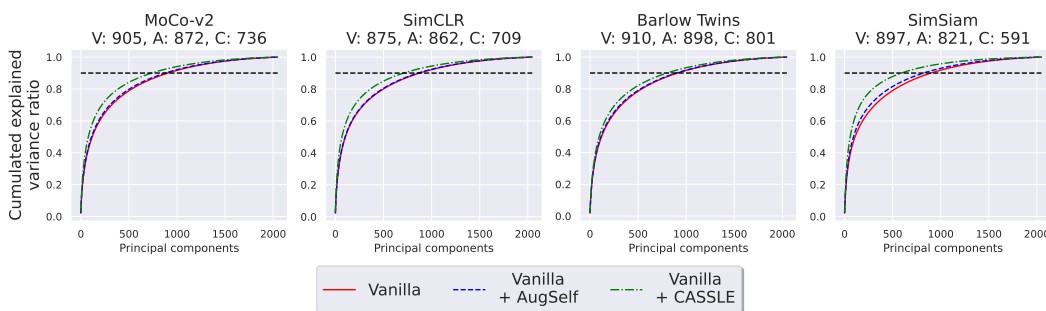

Figure 7: Cumulated explained variance ratio of representations learned by self-supervised models trained in their original forms, as well as with AugSelf (Lee et al., 2021) and CASSLE extensions. We additionally denote the number of principal components which that explain 90% of the variance of each representation. Models pretrained with CASSLE explain the largest amount of variance with the smallest amount of components, suggesting a larger dimensionality collapse of their representations.

**Analysis of the contrastive learning procedure** We next compare the training of MoCo-v2 (He et al., 2020; Chen et al., 2020b) with and without CASSLE or AugSelf (Lee et al., 2021) extensions, and plot the contrastive loss values measured throughout training in the left part of Figure 8, and on

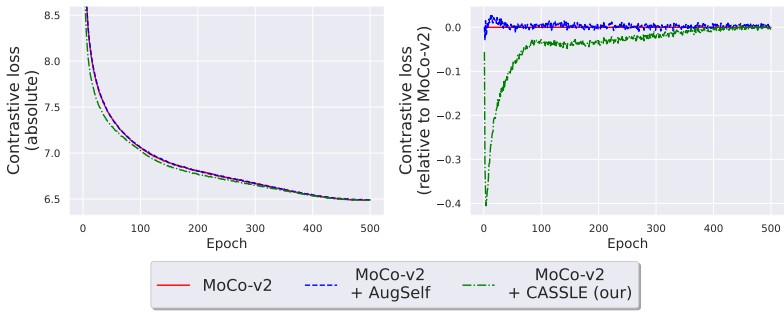

Figure 8: Absolute (left) and relative to Baseline (right) values of contrastive losses of Baseline, AugSelf (Lee et al., 2021), and CASSLE MoCo-v2 variants, measured during pretraining. CASSLE minimizes the contrastive objective faster than Baseline and AugSelf, in particular early in the training procedure.

the right, the values of losses relative to the baseline MoCo-v2. CASSLE minimizes the contrastive objective faster than the other two variants, in particular early in the training procedure. This suggests that augmentation information provides helpful guidance for a model not yet fully trained to align augmented image pairs and thus, CASSLE learns to depend on this information.

### C.3 EXPANDED ABLATION STUDY RESULTS

In order to optimally select the hyperparameters of CASSLE, we train different variants of MoCo-v2+CASSLE and evaluate them on the same classification and regression tasks as in Section 4.1 of the main paper. We rank the models from best to worst performance on each task and report the full results and average ranks in Table 12.

We recall the notation used for Augmentation information contents – $\omega^{\{x,y\}}$ denotes including parameters of augmentations $\{x, y\}$ in augmentation information vector $\omega$. For example, $\omega^{\{c,j\}}$ denotes $\omega$ containing **c**ropping and color **j**ittering parameters.

## D IMPLEMENTATION DETAILS

We implement CASSLE in the PyTorch framework (Paszke et al., 2019), building upon the codebase of (Lee et al., 2021). The anonymized version of our code is available at `https://anonymous.4open.science/r/CASSLE-037C/README.md`.

Table 12: Ablation study of CASSLE parameters. The results are computed with MoCo-v2+CASSLE. It is best to condition CASSLE on all available augmentation information. CASSLE yields the best results when implemented by concatenating or adding the augmentation and image embeddings together.

| Parameter | C10 | C100 | Food | MIT | Pets | Flowers | Caltech | Cars | FGVCA | DTD | SUN | CUB | 300W | Avg. rank ↓ |
|---|---|---|---|---|---|---|---|---|---|---|---|---|---|---|
| Augmentation information contents | | | | | | | | | | | | | | |
| $\omega^{\{c\}}$ | 84.89 | 62.95 | 59.74 | 63.96 | 72.26 | 83.55 | 79.66 | 38.78 | 42.03 | 65.11 | 48.44 | 33.86 | 89.17 | 4.54 |
| $\omega^{\{c,j\}}$ | 85.56 | 64.26 | 60.35 | 62.61 | 71.97 | 84.73 | 79.86 | 38.13 | 42.17 | 66.28 | 48.01 | 34.24 | 88.76 | 4.54 |
| $\omega^{\{c,j,d\}}$ | 85.87 | 63.91 | 61.07 | 63.51 | 72.71 | 86.53 | 79.51 | 38.27 | 42.53 | 66.70 | 49.27 | 35.76 | 89.11 | 3.08 |
| $\omega^{\{c,j,b,f\}}$ | 86.16 | 64.51 | 60.80 | 63.81 | 72.83 | 84.66 | 79.90 | 38.93 | 43.02 | 66.12 | 48.96 | 34.40 | 88.69 | 2.85 |
| $\omega^{\{c,j,b,f,g\}}$ | 85.85 | 64.14 | 61.24 | 63.73 | 72.88 | 84.50 | 79.93 | 38.23 | 41.28 | 65.27 | 48.90 | 34.47 | 88.78 | 3.54 |
| $\omega^{\{c,j,b,f,g,d\}}$ | 86.99 | 65.28 | 61.83 | 63.51 | 73.22 | 86.55 | 79.87 | 37.97 | 41.70 | 67.18 | 48.85 | 36.92 | 89.03 | **2.54** |
| Method of conditioning the projector | | | | | | | | | | | | | | |
| **Concatenation** | 86.99 | 65.28 | 61.83 | 63.51 | 73.22 | 86.55 | 79.87 | 37.97 | 41.70 | 67.18 | 48.85 | 36.92 | 89.03 | **1.92** |
| **Addition** | 86.45 | 65.40 | 63.00 | 65.15 | 71.34 | 86.91 | 79.79 | 37.83 | 42.18 | 66.17 | 49.28 | 37.42 | 88.87 | **1.92** |
| **Multiplication** | 86.72 | 66.70 | 60.65 | 60.97 | 64.60 | 85.17 | 80.09 | 33.54 | 41.56 | 63.99 | 47.63 | 32.15 | 89.48 | 2.69 |
| **Hypernetwork** | 84.70 | 63.55 | 60.62 | 64.10 | 67.16 | 82.76 | 78.47 | 33.39 | 39.85 | 66.44 | 47.43 | 30.48 | 89.11 | 3.46 |
| Depth of the Augmentation encoder (0 denotes concatenating raw $\omega$ to $\mathbf{e}$) | | | | | | | | | | | | | | |
| **0** | 86.53 | 65.99 | 62.54 | 61.72 | 69.04 | 85.46 | 80.74 | 36.44 | 41.91 | 65.64 | 48.55 | 33.88 | 92.72 | 3.38 |
| **2** | 86.57 | 64.80 | 61.85 | 62.68 | 72.79 | 86.31 | 79.93 | 37.85 | 42.87 | 66.32 | 49.23 | 35.39 | 88.86 | 3.08 |
| **4** | 86.32 | 65.16 | 61.98 | 64.93 | 72.64 | 86.49 | 79.75 | 38.64 | 41.46 | 66.91 | 49.71 | 36.56 | 89.27 | 2.38 |
| **6** | 86.99 | 65.28 | 61.83 | 63.51 | 73.22 | 86.55 | 79.87 | 37.97 | 41.70 | 67.18 | 48.85 | 36.92 | 89.03 | **2.15** |
| **8** | 85.47 | 64.88 | 61.49 | 63.13 | 72.41 | 85.65 | 78.22 | 37.82 | 40.71 | 66.70 | 49.21 | 36.09 | 88.90 | 4.00 |
| Hidden size of the Augmentation encoder | | | | | | | | | | | | | | |
| **16** | 84.51 | 63.40 | 61.38 | 62.31 | 71.61 | 85.40 | 78.96 | 37.53 | 41.84 | 66.54 | 48.68 | 35.54 | 88.78 | 5.54 |
| **32** | 85.43 | 63.94 | 61.93 | 64.18 | 72.05 | 85.67 | 79.80 | 37.86 | 41.52 | 67.13 | 48.71 | 35.64 | 88.67 | 4.38 |
| **64** | 86.99 | 65.28 | 61.83 | 63.51 | 73.22 | 86.55 | 79.87 | 37.97 | 41.70 | 67.18 | 48.85 | 36.92 | 89.03 | **2.54** |
| **128** | 86.24 | 64.66 | 61.95 | 64.25 | 72.12 | 86.72 | 79.56 | 37.79 | 42.71 | 67.61 | 49.52 | 36.87 | 88.99 | 2.62 |
| **256** | 86.23 | 65.63 | 61.77 | 61.79 | 72.03 | 85.69 | 80.38 | 37.86 | 40.94 | 67.07 | 49.59 | 37.00 | 89.37 | 3.31 |
| **512** | 85.77 | 66.05 | 62.21 | 64.55 | 72.45 | 86.38 | 80.06 | 37.91 | 41.94 | 66.22 | 49.35 | 36.24 | 88.87 | 2.69 |
| Impact of utilizing color difference information during pretraining | | | | | | | | | | | | | | |
| **AugSelf** $\omega^{\{c,j\}}$ | 85.26 | 63.90 | 60.78 | 63.36 | 73.46 | 85.70 | 78.93 | 37.35 | 39.47 | 66.22 | 48.52 | 37.00 | **89.49** | 3.07 |
| CASSLE $\omega^{\{c,j,b,f,g\}}$ | 85.85 | 64.14 | 61.24 | 63.73 | 72.88 | 84.50 | *79.93* | 38.23 | 41.28 | 65.27 | 48.90 | 34.47 | 88.78 | 2.64 |
| AugSelf $\omega^{\{c,j,d\}}$ | 84.95 | 64.06 | 61.53 | 63.06 | *73.52* | 86.25 | 77.38 | 36.00 | **42.54** | 66.33 | 48.65 | **37.40** | 88.36 | 2.71 |
| **CASSLE** $\omega^{\{c,j,b,f,g,d\}}$ | **86.32** | **65.29** | **61.93** | **63.86** | 72.86 | **86.51** | 79.63 | **38.82** | 42.03 | *66.54* | **49.25** | 36.22 | 88.93 | **1.57** |

