# OpenReview forum: "Augmentation-aware Self-Supervised Learning with Conditioned Projector"
_ICLR.cc/2024/Conference — Submitted to ICLR 2024_

### Official Review · Reviewer_Fj5t · 2023-10-31

**Soundness:** 2 fair
**Presentation:** 3 good
**Contribution:** 2 fair
**Rating:** 5
**Confidence:** 4

**Summary:**

This paper considers the problem of recent self-supervised learning methods that they learn to be invariant to data augmentations, which may be harmful for some downstream tasks. To tackle this problem, this paper proposes to modify the projector by feeding the information about data augmentations together with the encoder outputs. Experimental results show the effectiveness of the proposed method in transfer learning on image datasets.

**Strengths:**

- Learning augmentation-aware representations is a timely topic.

- The proposed idea is simple and ablation studies show how the design choices are made well.

**Weaknesses:**

- [Garrido et al.] would be one of the most recent work among prior works but missed in this paper.

- The proposed method simply provides the additional information about augmentations together with the encoder outputs, and it is not clear how it helps to "preserve more information about augmentations" in representations. Figure 3 shows that injecting information of random augmentations results in reduced cosine similarities. This implies that the projector relies on the given information about augmentations, which is not directly related to the learned representations (the output of the encoder), i.e., learned representations do not have to be changed regardless of whether the projector relies on the additional information about augmentations or not. Any theoretical justification on the effect of the proposed method to the learned representations would be welcome.

- The performance gain is overall minor and often it underperforms previous methods.

- Why does MoCo-v2 in Table 1 contain only one performance of LooC? It looks quite not informative.

- Why does MoCo-v3 in Table 1 miss the performance of "AI by [Chavhan et al.]," while the original paper presents its performance?

- The reference section requires thorough proofreading, as there are many incomplete/inaccurate references. For example, the closest prior work by [Lee et al.] is published in NeurIPS'21, but its arXiv version is cited. Also, many references miss the name of the published venue.

[Chavhan et al.] Amortised invariance learning for contrastive self-supervision. In ICLR, 2023.

[Garrido et al.] Self-supervised learning of Split Invariant Equivariant representations. In ICML, 2023.

**Questions:**

Please address concerns in Weaknesses.

> **post rebuttal**

As some of my concerns remain after discussion with authors, I keep my original rating unchanged with more confidence.

- I think your experiments can only be used to "indirectly" justify if the encoder preserves the information about augmentations. As I suggested, any theoretical analysis or a more direct experiment that checks if the proposed method better retains the augmentation information compared to the baseline would be helpful.

- Only ResNet-50 is used throughout experiments, so my concern on the scalability still remains. Note that ViT is just one option to resolve this concern; you can use different number of layers or other types of CNN architectures. Generally speaking, ViT becomes prevalent in the last ~ 2 years, so experiments with this type of architecture would strengthen your contribution.

- The reference section is still not proofread after being pointed out twice; at this point, I am not sure if authors are willing to show proper respect for previous works. MoCo v2 is an arXiv preprint and MoCo v3 is published in ICCV'21, but both miss their venues.

Xinlei Chen, Haoqi Fan, Ross Girshick, and Kaiming He. Improved baselines with momentum
contrastive learning, 2020b.

Xinlei Chen, Saining Xie, and Kaiming He. An empirical study of training self-supervised vision
transformers, October 2021b.

---

> ### Author Response · Authors · 2023-11-15
>
> Dear Reviewer Fj5t,
>
> Thank you vrey much for a thorough review of our work.  We would like to address your concerns below. If you have any additional questions, we are looking forward to answering them.
>
> **ad. W1: Comparison to the work of Garrido et. al.**
>
> Thank you for this question. In their recent work, Garrido et. al propose to extend VicReg with a hypernetwork-based predictor to learn representations that are equivariant to transformations. Similarly to one of the variants of CASSLE, their hypernetwork predicts the parameters of the predictor based on the transformation descriptors. In our work, we show that there exist methods of conditioning superior to hypernetworks. We also evaluate our approach on real-life image data, as well as a wider range of SSL methods.
>
> **ad. W2: request for additional theoretical justification:**
>
> We have expanded the discussion of Figure 3 in Section 3.2, showing that the conditional probability of feature extractor representations on the condition of correct augmentation information is larger than on the condition of any random augmentation information. This implies that the learned representations of images are correlated, to an extent, with the parameters of augmentations applied to them.
>
> **ad. W3: The performance gain is overall minor and often it underperforms previous methods**
>
> While it is true that CASSLE does not always achieve the best downstream performance, it does so in the majority of cases. We have compiled the linear evaluation results of different approaches on different downstream tasks (Tables 1 and 7) and ranked each approach from best to worst-performing in each downstream task (See Figure 5 in the revised manuscript).
>
> We find that:
>
> * CASSLE performs best in 54 out of 91 downstream tasks, i.e. 59.34% of cases where it was evaluated.
> * AugSelf performs best in 31 out of 91 downstream tasks, i.e. 34.07% of cases where it was evaluated.
> * AI performs best in 5 out of 16 downstream tasks, i.e. 31.25% of cases where it was evaluated.
> * IFM performs best in 1 out of 13 downstream tasks, i.e. 7.69% of cases where it was evaluated.
> * Vanilla performs best in 0 out of 91 downstream tasks, i.e. 0.00% of cases where it was evaluated.
>
> (we excluded LooC from the analysis, see the below answer to W4).
>
> **ad. W4: Why does MoCo-v2 in Table 1 contain only one performance of LooC? It looks quite not informative.**
>
> Out of the datasets we are transferring to in Table 1, the authors of LooC have conducted experiments only with transferring to the CUB dataset, which we re-reported in our work. To the best of our knowledge, neither the codebase nor network checkpoints trained by LooC are publicly available. As such, we were not able to benchmark this method on more data.
> It is also worth noting that both we and the authors of AugSelf observed lower performance of baseline MoCo-v2 transferred to CUB, compared to the authors of LooC.
> We also note that other works on augmentation-aware SSL [1,2] do not reimplement LooC and also re-reported their results.
>
> **ad. W5: Why does MoCo-v3 in Table 1 miss the performance of "AI by [Chavhan et al.]," while the original paper presents its performance?**
>
> This is because we trained a different backbone (ViT-Small, which is 4x smaller than ViT-base), compared to the one used by Chavhan et. al (ViT-Base).
>
> **ad. W6: updating the bibliography**
>
> Thank you for pointing this out. We have updated the bibliography in the revised version of the paper in order to refer to journal /conference versions of the cited papers.
>
>
> We hope that you find our answers satisfactory. We are looking forward to answering any of your additional questions.
>
> Kind regards,
>
> Authors
>
> [1] Amortised invariance learning for contrastive self-supervision. In ICLR, 2023.
>
> [2] Improving Transferability of Representations via Augmentation-Aware Self-Supervision Hankook Lee, Kibok Lee, Kimin Lee, Honglak Lee, Jinwoo Shin, 2021

---

> > ### Comment · Reviewer_Fj5t · 2023-11-22
> >
> > Thanks for addressing my questions. Below I provide more comments, that would hopefully be helpful regardless of the acceptance of this paper.
> >
> > > **ad. W2: request for additional theoretical justification:** We have expanded the discussion of Figure 3 in Section 3.2, showing that the conditional probability of feature extractor representations on the condition of correct augmentation information is larger than on the condition of any random augmentation information. This implies that the learned representations of images are correlated, to an extent, with the parameters of augmentations applied to them.
> >
> > I went through the revision, but still I don't think it fully explains if the representations $e$ retains the augmentation information. If there is no theoretical analysis to prove this, then an additional experiment to check if $e$ better retains the augmentation information when trained the proposed method compared to the baseline would be helpful.
> >
> > > **ad. W3: The performance gain is overall minor and often it underperforms previous methods** While it is true that CASSLE does not always achieve the best downstream performance, it does so in the majority of cases. We have compiled the linear evaluation results of different approaches on different downstream tasks (Tables 1 and 7) and ranked each approach from best to worst-performing in each downstream task (See Figure 5 in the revised manuscript).
> >
> > In my opinion, the proposed method does not have to outperform the others in all cases. Rather than counting how frequently the proposed method outperforms the others, analyzing why the proposed method fails to outperform the others would be more useful. By looking at Table 1 and 7, CASSLE is not better than AugSelf when combined with siamese representation learning (BYOL and SimSiam) and/or when the architecture is ViT.
> >
> > > **ad. W5: Why does MoCo-v3 in Table 1 miss the performance of "AI by [Chavhan et al.]," while the original paper presents its performance?** This is because we trained a different backbone (ViT-Small, which is 4x smaller than ViT-base), compared to the one used by Chavhan et. al (ViT-Base).
> >
> > Then, can you say the proposed CASSLE is scalable? Experimental results with a larger backbone would be preferable, as some progress on DL in small-scale settings is often not scalable.
> >
> > > **ad. W6: updating the bibliography** Thank you for pointing this out. We have updated the bibliography in the revised version of the paper in order to refer to journal /conference versions of the cited papers.
> >
> > I don't think they are properly fixed yet. I can see \citet is often misused in the place of \citep, and some references still missed their venue, e.g., CPC (Aaron van den Oord, Yazhe Li, and Oriol Vinyals. Representation learning with contrastive predictive coding, 2019.) is an arXiv paper, and the context prediction paper (Carl Doersch, Abhinav Gupta, and Alexei A. Efros. Unsupervised visual representation learning by context prediction. December 2015.) is published in ICCV'15. Also, references are generally too verbose.

---

> > > ### Author Response · Authors · 2023-11-22
> > >
> > > Dear Reviewer Fj5t,
> > >
> > > Thank you for your response and additional comments. We would like to answer them below.
> > >
> > > 1. We believe our paper sufficiently proves experimentally that CASSLE leads to preserving information about perturbed features.
> > >  - Figure 3 indicates that falsified augmentation information has a negative effect on CASSLE's projector performance. If the augmented features were not preserved in feature extractor representation, the projector would be indifferent to augmentation information used for conditioning.
> > >  - Figure 4 shows that CASSLE's representaion is more subject to augmentation-induced noise, compared to other models.
> > >  - Table 6 shows that, in case of SimCLR and Barlow Twins, CASSLE model performs best in the task of rotation prediction - prediction of augmentation which was not used during pretraining.
> > >
> > > 2. In our previous answer, we focused on answering the concern that "The performance gain is overall minor and often it underperforms previous methods.". We provided a measure that shows that CASSLE outperforms numerous other methods in the majority of cases. As for the discussion of BYOL and SimSiam, we suspect that this may be caused by the asymmetric representation used by those methods, which distill the representations between the projector and additional predictor network. Observe that CASSLE and AugSelf also perform comparably on MoCo-v3, which also utilizes an additional predictor. It is worth noticing that CASSLE improves the performance of the vanilla versions of BYOL and SimSiam as well. We have added this remark to the revised version of the paper.
> > >
> > > 3. Our work focuses primarily on convolutional architectures in augmentation-aware SSL. There are two main reasons for that: 1) convolutional architectures are the most popular ones in prior works on augmentation-aware SSL, such as AugSelf, AI and LooC; 2) following the main research line of augmentation-aware SSL we can fairly compare our methods and ideas against most of the existing methods using convolutional architectures.
> > > Please note that prior to our work, the only augmentation-aware method to have been benchmarked on ViTs was AI, where the ViT variant of the method worked because the authors trained it on top of pre-trained ViT. Moreover, in the case of ViT, their method was often outperformed by fine-tuning and not compared to AugSelf, leaving the question of scaling largely unanswered. On the other hand, we compare AugSelf and CASSLE with MoCo-v3, where all three models are trained from scratch. We believe that CASSLE should scale to larger models.
> > > While the question of whether augmentation-aware models scale to larger architectures is a valid one, it requires a more rigid analysis of all proposed techniques. For instance, larger models may benefit from different hyperparameter values of augmentation-aware methods than those established for ResNets. Such a detailed analysis is out of the scope of our paper.
> > >
> > > 4. In our manuscript we consistently use only the `\citep` command. We recognize that `\citet` would be preferable in a few places of the manuscript. We do not quite understand what you mean by "too verbose" references.  Thank you for pointing out those additional papers with missing metadata. We will take that into account in the revision of our paper.
> > >
> > > We are looking forward to answering any of your additional questions.
> > >
> > > Thank you,
> > >
> > > Authors

---

### Official Review · Reviewer_ad4G · 2023-11-01

**Soundness:** 2 fair
**Presentation:** 3 good
**Contribution:** 2 fair
**Rating:** 6
**Confidence:** 4

**Summary:**

Self-supervised methods are known to learn representations invariant to augmentations applied during training. This can be problematic when features of such augmentations are important for downstream tasks. This work considers the important task of performing self-supervised learning without losing important semantic features in the data. To achieve this, CASSLE is proposed, a method which conditions the learned projection head on the augmentations of each view. The work demonstrates that this results in features which are still augmentation-aware.

**Strengths:**

* The manuscript is well written and experiments are well picked to test the purported claims regarding sensitivity of learned features to augmentations applied during training.
* CASSLE is simple and has demonstrated efficacy when training augmentation-based contrastive models. When compared to other methods that condition on augmentations applied during training, table 1 shows that CASSLE has superior performance across many datasets.

**Weaknesses:**

* Based on Table 7, the proposed method seems to less effective for SimSiam and BYOL compared to InfoNCE based methods. The manuscript currently claims that CASSLE is applicable to all joint-embedding architectures, but the current experimental results do not demonstrate this.
* The experiments in 4.2 use the InfoNCE to evaluate augmentation-awareness, which is sensitive to the negative examples that are used. Instead of this, why not perform linear probing to predict the specific augmentation applied to an image? This would be a more direct measure of the augmentation-awareness.
* The work does not address the large body of work surrounding “feature suppression”, an important issue of contrastive models becoming invariant to features important for downstream tasks. I believe the work can be strengthened by including comparisons to methods proposed to address feature suppression [2], as well as evaluation on some feature suppression benchmarks [1].
* Current experiments do not demonstrate the effectiveness of CASSLE with augmentation-free approaches to self-supervised learning. This limits the modalities in which it can be applied to those where augmentations can be selected a priori.

Minor:
* For Table 1, and other similar tables, could the authors add a column denoting mean improvement, taken over datasets, over the vanilla baseline to more easily compare each of the methods to CASSLE? It does not have to specifically be an additional column, but it would be nice to have an aggregate metric of performance in comparison to the baseline.
* Some of the citations should be updated to include the full Author name (E.g., MoCo and SimSiam citations)


[1] "Intriguing Properties of Contrastive Losses,” Chen et al., 2021

[2] “Can contrastive learning avoid shortcut solutions?,” Robinson et al., 2021.

**Questions:**

* How does CASSLE relate to feature suppression [1] and shortcut solutions [2] in contrastive learning?
* Table 4 indicates that many of the methods were trained with a batch size of 256. Can the authors clarify why this was set so low? In the SimCLR paper it is shown that contrastive methods perform much worse when trained with a smaller batch size. Does CASSLE scale to larger batch sizes? Does CASSLE still perform well with a large batch size?
* Can CASSLE be applied to masked self-supervision? There seems to be a connection between CASSLE and the MAE, where the latter conditions on mask tokens to reconstruct masked patches.
* Have the authors tried performing feature inversion like in [3]? It would be interesting to see if CASSLE results in inverted features that are more reconstructive of attributes like color compared to vanilla contrastive features.


[1] "Intriguing Properties of Contrastive Losses,” Chen et al., 2021

[2] “Can contrastive learning avoid shortcut solutions?,” Robinson et al., 2021.

[3] “What makes instance discrimination good for transfer learning?,” Zhao et al., 2021.

---

> ### Author Response · Authors · 2023-11-15
>
> Dear Revier ad4G,
>
> Thank you for a detailed review of our work.  We would like to address your concerns below. If you have any additional questions, we are looking forward to answering them.
>
> **ad. W1: applicability to SimSiam and BYOL.**
>
> When talking about applicability, the meaning is that the design of CASSLE is compatible with a wide range of joint-embedding approaches and thus, can be applied to them. We do not claim that it is guaranteed to outperform approaches such as AugSelf. Nevertheless, while AugSelf outperforms CASSLE on SimSiam and BYOL, CASSLE still offers a performance boost compared to the vanilla versions of those methods, confirming its applicability in the sense of performance.
>
>
> **ad. W2: predicting augmentation parameters instead of InfoNCE in 4.2**
>
> Thank you for your suggestion. We do not directly predict the augmentation parameters as we find it quite inconsequential. During training, we present the perturbed images with the induced parameters so that we build a stronger feature extractor, informed by the nuances caused by the augmentations. Note that for Augself it makes sense to predict the (difference of) augmentations as it is the key component for their loss function. However, in this experiment, we aim to measure the sensitivity in various stages rather than a quantitative description of applied augmentations.
>
>
> **ad. W4: comparison with augmentation-free approaches**
>
> Our research problem focuses on how augmentation-based SSL methods become invariant to augmentations. We are not aware of any evidence in the literature that MIM-based methods also suffer from this issue. As such, a comparison to them is not relevant in the context of CASSLE.
>
> **ad. W5: aggregate metric of improvement over baseline**
>
> Thank you for this valuable suggestion. We have compiled the linear evaluation results of different approaches on different downstream tasks (Tables 1 and 7) and summarized them in Figure 5 (appendix C) in the revised manuscript. CASSLE is usually ranked the best in terms of performance on downstream tasks compared to AugSe and Vanilla approaches. AugSelf and CASSLE improve over Vanilla approaches by a comparable margin. Finally, CASSLE achieves the best performance in the largest number of downstream tasks.
>
> **ad. W7: updating the bibliography**
>
> Thank you for raising this concern. We have updated the bibliography in the revised version of the paper in order to refer to journal and conference versions of the cited papers.
>
> **ad. Q1/ W3: How does CASSLE relate to feature suppression [1] and shortcut solutions [2] in contrastive learning?**
>
> Thank you for pointing this out. We believe that invariance to augmentation can be regarded as a consequence of feature suppression, and have added an according reference to the above works in the Related Work section of the revised version of our manuscript. It is also interesting to verify how transferable are methods designed for mitigating feature suppression. We also added a comparison to MoCo-v2 + implicit feature modification (IFM) [2] to find how this established method of mitigating feature suppression transfers to downstream tasks.
>
> **ad. Q2: Question about batch size of 256**
>
> We have set this batch size for all methods in compliance with the hyperparameters of AugSelf [4]. We note that the batch size is mainly important for SimCLR, as approaches such as MoCo decouple the batch size from the number of negatives, and Barlow Twins, SimSiam and BYOL work well with batch sizes of 256. CASSLE is unrelated to the batch sizes and we expect it to perform equally well with larger batch sizes.
>
> **ad. Q3: Can CASSLE be applied to masked self-supervision?**
>
> Indeed, there seems to be such a connection. While CASSLE has been designed with joint-embedding approaches in mind, applying it to MIM-based approaches could be an interesting idea for future work. However, we are not aware of any works that indicate that MIM-based methods also suffer from the issue of augmentation invariance.
>
> **ad. Q4: Have the authors tried performing feature inversion like in [3]?**
>
> While this could be interesting, we did not perform such an evaluation. We are interested in whether features learned by CASSLE are more transferable to downstream tasks and harder to match together, whereas the quality of reconstruction of particular attributes such as color could be subject to large variance.
>
> We hope that you find our answers satisfactory. We are looking forward to answering any of your additional questions.
>
> Kind regards,
>
> Authors
>
> [1] "Intriguing Properties of Contrastive Losses,” Chen et al., 2021
>
> [2] “Can contrastive learning avoid shortcut solutions?,” Robinson et al., 2021.
>
> [3] “What makes instance discrimination good for transfer learning?,” Zhao et al., 2021.
>
> [4] Improving Transferability of Representations via Augmentation-Aware Self-Supervision Hankook Lee, Kibok Lee, Kimin Lee, Honglak Lee, Jinwoo Shin, 2021

---

> > ### Comment · Reviewer_ad4G · 2023-11-19
> > **Response to Rebuttal**
> >
> > I thank the authors for engaging with the review of their work. I believe the additional baseline comparisons and discussion of related works strengthen the work, and I have updated my score as such.
> >
> > I disagree that predicting the augmentations applied to an image would be inconsequential to support the claim that CASSLE reduces feature-space invariance to the augmentations used during training. I view the use of the InfoNCE objective to measure this as problematic, since this loss is sensitive to the choice of negative examples and the hyper-parameters selected (making it difficult to compare between different methods).

---

> > > ### Author Response · Authors · 2023-11-20
> > > **Thank you**
> > >
> > > Dear Reviewer ad4G,
> > >
> > > We are grateful for your reply and for upgrading your score. We have two additional comments.
> > >
> > > --------
> > > 1. Regarding the discussion of the invariance experiment, we would like to clarify that in the experiment presented in Section 4.2, we measure the InfoNCE of pretrained models in the inference mode. Let us describe this in a greater detail:
> > >
> > > * for all pretrained networks, we process through them the ImageNet-100 test dataset. We infer the representations of subsequent ResNet blocks and their projector networks.
> > > * for representations acquired from each stage of the network, we measure the value of InfoNCE, i.e. the cross-entropy of correctly matching embeddings of two augmented image views vs. all the images in the batch (we use the batch size of 256).
> > > * in the case of CASSLE, we additionally supply the information about applied augmentations to the projector (just like during pretraining)
> > > * all other parameters of the experiment are identical between models.
> > >
> > > As such, for each model, we measure the degree to which augmentations render the model's representations of positive image pairs unmatchable among representations of other images in the given batch. We believe this constitutes a fair comparison.
> > >
> > > --------
> > > 2. In order to fully answer your question about the batch size in SimCLR, we ran experiments with this batch size for vanilla SimCLR, AugSelf, and CASSLE. We present the linear evaluation in the table below.
> > >
> > > | SimCLR   | C10       | C100      | Food      | MIT       | Pets      | Flowers   | Caltech   | Cars      | FGVCA     | DTD       | SUN       | CUB       | 300W      |
> > > |----------|-----------|-----------|-----------|-----------|-----------|-----------|-----------|-----------|-----------|-----------|-----------|-----------|-----------|
> > > | Vanilla  |     84.41 |     61.77 |     57.48 |     63.10 |     71.60 |     83.37 | **79.67** |     35.14 |     40.03 |     65.59 |     46.92 |     30.98 |     88.59 |
> > > | AI       |     83.90 |     63.10 | --        | --        |     69.50 |     68.30 |     74.20 | --        | --        |     53.70 | --        | **38.60** |     88.00 |
> > > | AugSelf  |     84.45 |     62.67 |     59.96 |     63.21 |     70.61 | **85.77** |     77.78 |     37.38 |     42.86 |     65.53 | **49.18** |     34.24 |     88.27 |
> > > | CASSLE   | **86.31** | **64.36** | **60.67** | **63.96** | **72.33** |     85.22 |     79.62 | **39.86** | **43.10** | **65.96** |     48.91 |     33.21 | **88.88** |
> > >
> > > We observe that while most of the results are indeed better than for SimCLR variants trained with the batch size of 256, the relative downstream performance of the compared extensions remains relatively unchanged. We maintain that CASSLE is the best-performing extension of SimCLR among the compared methods.
> > >
> > > However, a full benchmark of the new SimCLR variants would require us to re-run all the evaluations reported in our manuscript, in particular the lengthy object detection evaluation, which we do not have the capacity for within the remaining discussion timeframe. In order to not cause confusion about which SimCLR model was used in which evaluation, we choose to keep the original results in the manuscript for now. We will conduct a full evaluation of the models trained with larger batch sizes for the camera-ready version.
> > >
> > > Thank you once again for your constructive feedback,
> > >
> > > Authors

---

### Official Review · Reviewer_N37y · 2023-11-01

**Soundness:** 3 good
**Presentation:** 4 excellent
**Contribution:** 3 good
**Rating:** 6
**Confidence:** 4

**Summary:**

State-of-the-art approaches to self-supervised representation learning (SSL) optimize invariance-inducing objectives of representations to augmented views of an input observation, while preventing their collapse to a trivial solution. Effective optimization of these objectives reasonably results to information loss about the features excited by the augmentations in the representation space. These features, however, could be useful to maintain for some downstream prediction (potentially transfer learning) tasks. While the projector network is a common feature of these methods which mitigates this effect, the invariance still persists. The authors propose a simple intervention to typical SSL pipelines in order to further mitigate this effect: ***they suggest to condition the projector network with information about the particular augmentation used to derive a view of an observation***. Experiments on downstream transfer learning tasks with pretrained networks demonstrate an improvement in performance compared to baseline methods, targeted at the same issue. Analysis of representations and the projector demonstrate that augmentation information is indeed used and the method leads to more sensitivity to the variations induced by augmentations in earlier activations of the pretrained network. They also provide with ablation analyses of various implementation design choices.

**Strengths:**

1. The identified problem is known and significant for representation learning. The authors discuss fairly well the related literature and approaches to its solution.
2. The idea is fairly novel, there have been some similar approaches that essentially “condition the projector network”. Please, refer to Question 1.
3. Nonetheless, their results generally convince that the detail is in the implementation level, rather than the conceptual.
4. The paper is well-written and well-argumented.

Overall, the paper convinces that conditioning the projector with augmentation information is a good direction towards creating more potent and transferable representations.

**Weaknesses:**

1. Experiments remain relatively small-scale in dataset and model size. Especially, it would have been interesting to examine the effect of conditioning as pretraining data becomes abundant.
2. CASSLE performs better (compared to AugSelf) for contrastive methods and BarlowTwins than others, i.e. BYOL and SimSiam. A discussion on why this happens can be interesting.
3. Semi-supervised (few-shot classification) results are competitive, but weaker.
4. Experiments on object detection task demonstrate a marginal improvement.

5. The paper does not report confidence intervals of their results.
6. Citation and bibliography style needs serious editing. Sometimes “et al.” is retained in bibliography, journals/conferences/proceedings are frequently missing and style is generally inconsistent.

**Questions:**

1. Missing relevant approach to CASSLE is [1]. They provide with a method which can be perceived as a kind of conditioning to augmentation information.
2. In *Related Work*, contrastive learning objectives usually refer to methods which prevent representational collapse by contrasting against negative pairs. Please clarify this distinction.
3. In *Section 4.2*, an analysis of activation invariance is presented based on the InfoNCE loss. Which similarity function was used to compare earlier representations?
4. In *Table 3*, how is the rank computed exactly?

[1] Bhardwaj, Sangnie, et al. "Steerable equivariant representation learning." arXiv preprint arXiv:2302.11349 (2023).

---

> ### Author Response · Authors · 2023-11-15
>
> Dear Reviewer N37y,
>
> Thank you for your positive review of our work and all the comments. We would like to address your concerns below. If you have any additional questions, we are looking forward to answering them.
>
>
> **ad. W2: CASSLE performs better (compared to AugSelf) for contrastive methods and BarlowTwins than others, i.e. BYOL and SimSiam. A discussion on why this happens can be interesting.**
>
> Thank you for this question. We suspect that this may be caused by the asymmetric representation used by BYOL and SimSiam, which distill the representations between the projector and additional predictor network. Observe that CASSLE and AugSelf also perform comparably on MoCo-v3, which also utilizes an additional predictor. It is worth noticing that CASSLE improves the performance of the vanilla versions of those methods as well.
> We have added this remark to the revised version of the paper.
>
> **ad. W4: Experiments on object detection task demonstrate a marginal improvement.**
>
> While this improvement is small, CASSLE consistently improves over the performance of Vanilla and AugSelf models for both MoCo-v2 and SimCLR.
>
> **ad. W5: The paper does not report confidence intervals of their results.**
>
> We did not report the confidence intervals of individual downstream task performance, as this would limit the readability of (already massive) Tables 1 and 7. We have compiled the linear evaluation results of different approaches on different downstream tasks in Figure 5 (Appendix C) of the revised manuscript and report the mean rank of each model (when ranked from worst to best performance on downstream tasks) and mean improvements of CASSLE and AugSelf over Vanilla approaches, both with 95% confidence intervals.
>
> **ad. W6: Citation and bibliography style needs serious editing. **
>
> Thank you for pointing this out. We have updated the bibliography in the revised version of the paper in order to refer to journal /conference versions of the cited papers.
>
> **ad. Q1: Comparison to Bhardwaj et. al. [1]**
>
> The authors of [1] use a mapping function that, similarly to our conditioned projector, acts on the combined information of image embeddings and augmentation parameters. This mapping allows for supplementing the training objective with equivariance regularization and leads to learning representations that are equivariant to augmentations applied to the images. There are two key differences between our work and [1]:
> CASSLE does not modify the objective function of the trained model. While we do not optimize for equivariance, we observe that increased sensitivity to augmentations emerges in CASSLE due to injecting augmentation information into the projector.
> [1] is evaluated in the supervised learning setting, whereas CASSLE is designed for self-supervised learning. We have included a reference to [1] in our related work section.
>
> **ad. Q2: Contrastive learning objectives nomenclature**
>
> While there obviously exists a distinction between contrastive, distillation-based, and CCA-based approaches, their objectives are often collectively referred to as contrastive objectives [2,3]. We have adopted this term in our work since CASSLE is applicable to different joint-embedding models regardless of their objective. We are open to your suggestions for a more accurate term describing those three kinds of objectives.
>
> **ad. Q3: Which similarity function was used to compare earlier representations in Section 4.2?**
>
> For all stages of the network, we measure the cosine similarity of representations and calculate the InfoNCE loss subsequently.
>
>
> **ad. Q4: In Table 3, how is the rank computed exactly?**
>
> We compare different variants of MoCo-v2+CASSLE on the same classification and regression tasks as in Section 4.1. We rank the models from best to worst performance on each task and report the average ranks in Table 3. The full results are in Table 12.
>
> We hope that you find our answers satisfactory. We are looking forward to answering any of your additional questions.
>
> Kind regards,
> Authors

---

> > ### Comment · Reviewer_N37y · 2023-11-22
> > **Answer to rebuttal**
> >
> > Thank you for your responses! I choose to maintain my original positive assessment of the paper!

---

> > > ### Author Response · Authors · 2023-11-22
> > > **Thank you**
> > >
> > > Dear Reviewer N37y,
> > >
> > > Thank you once again for your positive assessment of our work and all the feedback.
> > >
> > > Kind regards,
> > >
> > > Authors

---

### Official Review · Reviewer_hzhB · 2023-11-01

**Soundness:** 3 good
**Presentation:** 3 good
**Contribution:** 2 fair
**Rating:** 3
**Confidence:** 5

**Summary:**

Many self-supervised learning methods aim to learn augmentation-invariant representations. Such an approach could be harmful when a downstream task is sensitive to augmentation-aware information. To overcome this limitation of existing SSL methods, this paper proposes a simple yet effective approach that injects augmentation information (i.e., augmentation parameters) into the projection MLP used in the SSL framework. The approach shows superior performance over existing augmentation-aware information learning methods on ImageNet-100 experiments.

**Strengths:**

- This paper is generally well-written. It is easy to understand.
- The idea is simple, intuitive, and seems to be widely applicable.
- The proposed method, CASSLE, outperforms baselines (LooC, AugSelf, and AI) that also learn augmentation-aware information.

**Weaknesses:**

**(1) Lack of comparison with recent augmentation-free SSL methods.** \
Recently, there have been proposed many augmentation-free self-supervised learning methods, including data2vec [1-2], I-JEPA [3], and Masked Image Modeling (MIM) [4-5]. The augmentation-free SSL methods do not use augmentation, in other words, they aim to learn full information about original images, rather than learning augmentation-invariant representations. Also, since they are often better than MoCo-v2 and SimCLR in various benchmarks (e.g., linear evaluation, fine-tuning, scalability), the authors should compare the proposed method with the methods.

[1] Baevski et al., data2vec: A General Framework for Self-supervised Learning in Speech, Vision and Language, ICML 2022 \
[2] Baevski et al., Efficient Self-supervised Learning with Contextualized Target Representations for Vision, Speech and Language, 2022 \
[3] Assran et al., Self-Supervised Learning from Images with a Joint-Embedding Predictive Architecture, ICCV 2023 \
[4] He et al., Masked Autoencoders Are Scalable Vision Learners, CVPR 2022 \
[5] Xie et al., SimMIM: a Simple Framework for Masked Image Modeling, CVPR 2022

**(2) Experimental results are not convincing.** \
The performance improvement of CASSLE over AugSelf is marginal.

**(3) Lack of novelty.** \
I feel that the proposed method is neither novel nor interesting. First, the goal of this paper has been widely studied via augmentation-aware objectives (e.g., AugSelf) and augmentation-free SSL methods (e.g., I-JEPA). Also, it is hard to find a strong advantage of the proposed idea compared to AugSelf. In my opinion, the choice between injection and prediction cannot make meaningful novelty.

**Questions:**

Can the proposed method be applied to generative modeling like GAN training? It is worth noting that the main baseline, AugSelf, can be utilized for efficient GAN training [1].

[1] Hou et al., Augmentation-Aware Self-Supervision for Data-Efficient GAN Training, NeurIPS 2023

---

> ### Author Response · Authors · 2023-11-15
> **Reply (1/2)**
>
> Dear Reviewer hzhB,
>
> Thank you for a detailed review of our work. We would like to address your concerns below. If you have any additional questions, we are looking forward to answering them. Please note that we have split our answer into two comments.
>
> **ad. W1: Lack of comparison with recent augmentation-free SSL methods.**
>
> Our research problem focuses on how augmentation-based SSL methods become invariant to augmentations. We are not aware of any evidence in the literature that MIM-based methods also suffer from this issue. As such, a comparison to them is not relevant in the context of CASSLE.
>
> **ad. W2: The performance improvement of CASSLE over AugSelf is marginal.**
>
> While it is true that CASSLE does not always achieve the best downstream performance, it does so in the majority of cases. We have compiled the linear evaluation results of different approaches on different downstream tasks (Tables 1 and 7) and ranked each approach from best to worst-performing in each downstream task (See Figure 5 in the revised manuscript). We find that:
>
> * CASSLE performs best in 54 out of 91 downstream tasks, i.e. 59.34% of cases where it was evaluated.
> * AugSelf performs best in 31 out of 91 downstream tasks, i.e. 34.07% of cases where it was evaluated.
> * AI performs best in 5 out of 16 downstream tasks, i.e. 31.25% of cases where it was evaluated.
> * IFM performs best in 1 out of 13 downstream tasks, i.e. 7.69% of cases where it was evaluated.
> * Vanilla performs best in 0 out of 91 downstream tasks, i.e. 0.00% of cases where it was evaluated.
>
>
> **ad. W3a: Lack of novelty. First, the goal of this paper has been widely studied via augmentation-aware objectives (e.g., AugSelf) and augmentation-free SSL methods (e.g., I-JEPA).**
>
> We disagree that the existence of augmentation-free methods such as MIM and I-JEPA exhausts the need for augmentation-based methods and therefore, for research on augmentation awareness. We see several reasons for research into augmentation-aware approaches:
> * recent augmentation-based SSL approaches such as DINO-v2 [9] achieve state-of-the-art against augmentation-free methods
> * augmentation-based and augmentation-free methods learn different kinds of features - while augmentation-based methods lack spatial sensitivity which requires modeling the local structure within each image, MIM does not have good semantic alignment [8].
> * it should also be highlighted that augmentation-free models focus primarily on pretraining Vision Transformers (with some rare exceptions [7]). On the other hand, contrastive SSL continues to be the mainstream family of approaches for pretraining convolutional architectures such as ResNets, which continue to be widely used.
>
> Thus, better understanding and improving these methods remains an important research problem.
>
>
> [1] Baevski et al., data2vec: A General Framework for Self-supervised Learning in Speech, Vision and Language, ICML 2022
>
> [2] Baevski et al., Efficient Self-supervised Learning with Contextualized Target Representations for Vision, Speech and Language, 2022
>
> [3] Assran et al., Self-Supervised Learning from Images with a Joint-Embedding Predictive Architecture, ICCV 2023
>
> [4] He et al., Masked Autoencoders Are Scalable Vision Learners, CVPR 2022
>
> [5] Xie et al., SimMIM: a Simple Framework for Masked Image Modeling, CVPR 2022
>
> [6] What Should Not Be Contrastive in Contrastive Learning Tete Xiao, Xiaolong Wang, Alexei A. Efros, Trevor Darrell
>
> [7] Designing BERT for Convolutional Networks: Sparse and Hierarchical Masked Modeling Keyu Tian, Yi Jiang, Qishuai Diao, Chen Lin, Liwei Wang, Zehuan Yuan
>
> [8] Siamese Image Modeling for Self-Supervised Vision Representation Learning Chenxin Tao, Xizhou Zhu, Weijie Su, Gao Huang, Bin Li, Jie Zhou, Yu Qiao, Xiaogang Wang, Jifeng Dai; Proceedings of the IEEE/CVF Conference on Computer Vision and Pattern Recognition (CVPR), 2023, pp. 2132-2141
>
> [9] DINOv2: Learning Robust Visual Features without Supervision https://arxiv.org/abs/2304.07193
>
> [10] Hou et al., Augmentation-Aware Self-Supervision for Data-Efficient GAN Training, NeurIPS 2023

---

> ### Author Response · Authors · 2023-11-15
> **Reply (2/2)**
>
> **Note**: This is the second part of our reply.
>
> **ad. W3b: Also, it is hard to find a strong advantage of the proposed idea compared to AugSelf.**
>
> We believe our approach has several advantages over AugSelf:
> * a stronger theoretical justification, which we added in the revised Section 3.2 of the paper
> * AugSelf is a multi-task learning method. It requires the user to choose the proportions of the SSL, and augmentation prediction losses. In fact, the authors of AugSelf show that different proportions are needed for different types of backbones and that the choice of augmentations whose parameters to predict during pretraining is not trivial.
> * On the other hand, CASSLE does not introduce new losses to the training. We also demonstrate that utilizing all possible augmentation information during pretraining yields the best downstream performance.
> * Finally, during linear evaluation, there are 12 cases of AugSelf performing worse than the vanilla approach (in MoCo-v1, MoCo-v3, SimCLR, SimSiam, Barlow Twins), whereas CASSLE does so in only 4 cases which are isolated to the BYOL model.
>
> Thus, CASSLE requires simpler modifications to the extended SSL models, while offering a larger performance boost than AugSelf in the majority of cases.
>
>
>
> ad. Q1: Can the proposed method be applied to generative modeling like GAN training?
>
> Certainly, CASSLE could be used for efficient GAN training in a similar manner to AugSelf as shown in [10]. In this case, we should combine CASSLE conditioning with a discriminator network. Verification of this approach is however out of the scope of this paper but seems to be an interesting future direction.
>
> We hope that you find our answers satisfactory. We are looking forward to answering any of your additional questions.
>
> Kind regards,
>
> Authors
>
> [1] Baevski et al., data2vec: A General Framework for Self-supervised Learning in Speech, Vision and Language, ICML 2022
>
> [2] Baevski et al., Efficient Self-supervised Learning with Contextualized Target Representations for Vision, Speech and Language, 2022
>
> [3] Assran et al., Self-Supervised Learning from Images with a Joint-Embedding Predictive Architecture, ICCV 2023
>
> [4] He et al., Masked Autoencoders Are Scalable Vision Learners, CVPR 2022
>
> [5] Xie et al., SimMIM: a Simple Framework for Masked Image Modeling, CVPR 2022
>
> [6] What Should Not Be Contrastive in Contrastive Learning Tete Xiao, Xiaolong Wang, Alexei A. Efros, Trevor Darrell
>
> [7] Designing BERT for Convolutional Networks: Sparse and Hierarchical Masked Modeling Keyu Tian, Yi Jiang, Qishuai Diao, Chen Lin, Liwei Wang, Zehuan Yuan
>
> [8] Siamese Image Modeling for Self-Supervised Vision Representation Learning Chenxin Tao, Xizhou Zhu, Weijie Su, Gao Huang, Bin Li, Jie Zhou, Yu Qiao, Xiaogang Wang, Jifeng Dai; Proceedings of the IEEE/CVF Conference on Computer Vision and Pattern Recognition (CVPR), 2023, pp. 2132-2141
>
> [9] DINOv2: Learning Robust Visual Features without Supervision https://arxiv.org/abs/2304.07193
>
> [10] Hou et al., Augmentation-Aware Self-Supervision for Data-Efficient GAN Training, NeurIPS 2023

---

> ### Comment · Reviewer_hzhB · 2023-11-23
> **Thank you for the response.**
>
> Thank you for your efforts and time in this response.
>
> I have thoroughly read the response, but I remain unconvinced of the effectiveness of the proposed method for representation learning; hence, I will maintain my score.
>
> **(W1) Necessity of comparison with augmentation-free methods is required.** I believe all self-supervised learning (SSL) methods share the common goal of "learning better representations from unlabeled images". In this context, CASSLE employs augmentation-aware information while the augmentation-free methods (e.g., MIM) take alternative approaches. From my perspective, "learning augmentation-aware information" appears to be an approach rather than a distinct problem. Since CASSLE and other SSL methods are addressing the same problem, a comparison with augmentation-free methods should be provided.
>
> **(W2) Marginal performance improvements.** I think the winning rate of ~50% with a marginal gap is not convincing.
>
> **(W3) Lack of novelty.** While I acknowledge the need for research on augmentation-aware approaches, this cannot be the reason of the unnecessary of the comparison with recent (augmentation-free) methods as previously mentioned. Also, there have been proposed many SSL methods to extract more (e.g., augmentation-aware) information (e.g., patch-level objective in iBOT and multi-crop in DINO) as well as augmentation-free methods (MIM, I-JEPA, ...). Compared to the recent progress in the SSL literature, I think that "the proposed method is superior to AugSelf" is insufficient. In addition, although I acknowledge several advantages of CASSLE over AugSelf, but the advantages seem not strong due to the unsatisfactory performance improvements.

---

> > ### Author Response · Authors · 2023-11-23
> >
> > Dear Reviewer hzhB,
> >
> > thank you for your response and feedback. We would like to answer your comments below:
> >
> > **ad. 1 .** The subject of our paper is the augmentation-awareness of joint-embedding SSL methods, and it falls into a line of work covered extensively in the literature [1, 2, 3, 4]. None of these prior works compare their proposed methods to augmentation-free frameworks. Such a comparison would be a non-informative one -- downstream performance of models is influenced by a lot of factors, including sizes of the models, sizes of the pretraining datasets, SSL objective, etc. Our methodology is to compare augmentation-aware approaches on each SSL framework separately to isolate the difference in performance due to differing augmentation-aware approaches. Otherwise, we would be comparing apples to oranges. Note that our paper focuses primarily on convolutional architectures, to which most of the mentioned augmentation-free methods are not applicable. What metric would you suggest to fairly compare augmentation-aware and augmentation-free methods with different architectures?
> >
> > **ad. 2.** CASSLE outperforms other methods in nearly 60% of cases overall. For individual models, those numbers are:
> >
> > * MoCo-v1 - 100%
> > * MoCo-v2 - 69%
> > * MoCo-v3 - 54%
> > * SimCLR - 69%
> > * BYOL - 23%
> > * SimSiam - 8%
> > * Barlow Twins - 92%
> >
> > In other words, cases, where CASSLE is not the best method, are mostly isolated to BYOL and SimSiam, whereas for other methods, CASSLE is consistently the best-performing extension.
> >
> > **ad. 3.**  While we agree that augmentation-free methods may not have the augmentation-awareness problem, for reasons discussed in ad. 1, we feel that comparison to them would bring little information from the perspective of CASSLE. As for performance improvements over AugSelf, it performs best only for Distillation-based frameworks, whereas CASSLE is the superior method for contrastive-based and correlation-based frameworks. Also, CASSLE has a stronger theoretical foundation. We believe this constitutes a plausible performance improvement.
> >
> > Thank you once again for the discussion,
> >
> > Authors
> >
> > [1] What Should Not Be Contrastive in Contrastive Learning Tete Xiao, Xiaolong Wang, Alexei A. Efros, Trevor Darrell
> >
> > [2] Improving Transferability of Representations via Augmentation-Aware Self-Supervision Hankook Lee, Kibok Lee, Kimin Lee, Honglak Lee, Jinwoo Shin, 2021
> >
> > [3] Amortised invariance learning for contrastive self-supervision. In ICLR, 2023.
> >
> > [4] Self-supervised learning of Split Invariant Equivariant representations, Quentin Garrido, Laurent Najman, Yann LeCun, ICML 2023

---

### Author Response · Authors · 2023-11-15

Dear Reviewers and Area Chairs,

We thank all reviewers for taking the time to review our work and we really appreciate receiving numerous valuable feedback and diverse remarks which help us to strengthen the quality of our work.

The reviewers find our proposed method intuitive, well-motivated, fairly novel, and widely applicable, note that it outperforms numerous augmentation-aware baselines across many datasets, and compliment our design choices made in the ablation study. According to the reviewers, our paper identifies a known and significant problem in representation learning, discusses the related literature adequately, as well as picks good experiments to verify the performance of the discussed approaches. We are also delighted to hear that the presentation of our work is excellent and that our paper is well-written.

Based on the reviews, we made the following changes to the paper for the revised version.

* We include a more comprehensive, aggregated comparison of the downstream performance of the compared approaches. We have compiled the linear evaluation results of different approaches on different downstream tasks (Tables 1 and 7), ranked each approach from best to worst-performing in each downstream task, and added a new Figure 5 with bar plots displaying this data. The results show that CASSLE is, on average, the best-performing method for the majority of extended SSL approaches. (hzhB, N37y, ad4G, Fj5t)
* We have revised our bibliography section to cite the appropriate venues where the cited papers were published. (N37y, ad4G, Fj5t)
* We have expanded the related work section with a discussion of several works highlighted by the Reviewers. (N37y, ad4G, Fj5t)
* We added a comparison of MoCo-v2 to the IFM technique for preventing feature suppression. (ad4G)
* We added an additional theoretical discussion of the experiment presented in Figure 3. (Fj5t)


We provide the answers to every reviewer in responses directly to their comments. We hope that our responses and new results will answer your questions related to our work. If not, per your request we will do our best to follow up with additional results during the discussion period.

We are very much looking to engage in discussions with you and answer any of your additional questions.

Sincerely,

Authors

---

### Author Response · Authors · 2023-11-22
**A gentle reminder**

Dear Reviewers,

We would like to thank you once again for taking the time to review our work. We kindly remind you that we are less than a day from the end of the discussion period. We hope that we successfully addressed your questions.

If you have any further questions, please let us know - we are looking forward to discussing them with you.

Thank you very much!

Authors

---

### Meta-Review · Area_Chair_jYNQ · 2023-12-08

**Metareview:**

The paper introduces CASSLE, a novel approach in self-supervised learning (SSL) that conditions the projector on augmentation information, aiming to retain augmentation-sensitive features for downstream tasks. This approach addresses a crucial limitation in current SSL methods that often learn augmentation-invariant representations, potentially detrimental to certain downstream applications.

The reviewers agree on the paper's well-written nature and the relevance of the topic. The proposed method, CASSLE, is acknowledged for its novelty and potential in creating more potent and transferable representations. However, there are significant concerns regarding the novelty, scale of experiments, and comparison with state-of-the-art methods. The reviewers also express the need for more comprehensive analysis and theoretical justification.

The paper's primary strength lies in addressing a critical limitation in SSL methods by introducing augmentation-awareness, which is a timely and relevant topic. However, the concerns about its novelty, experimental scope, and theoretical grounding suggest that the paper might benefit from further development and a more comprehensive evaluation. Additionally, addressing the concerns regarding scalability and detailed comparisons with both contrastive and augmentation-free SSL methods could strengthen the paper's contribution to the field.

In conclusion, while the paper presents a promising approach, the mixed reviews and identified gaps indicate that it may require additional work to solidify its contributions and address the highlighted concerns fully.

**Justification For Why Not Higher Score:**

As said in meta review

**Justification For Why Not Lower Score:**

NA

---

### Decision · Program_Chairs · 2024-01-16

Reject